# VEGA: Learning Interleaved Image-Text Comprehension in Vision-Language Large Models

**Chenyu Zhou**[1,*]**, Mengdan Zhang**[*]**, Peixian Chen**[♠,*]**, Chaoyou Fu, Yunhang Shen**
**Xiawu Zheng**[1,†]**, Xing Sun, Rongrong Ji**[1]

[1]Key Laboratory of Multimedia Trusted Perception and Efficient Computing,
Ministry of Education of China, Xiamen University

## ABSTRACT

The swift progress of Multi-modal Large Language Models (MLLMs) has showcased their impressive ability to tackle tasks blending vision and language. Yet, most current models and benchmarks cater to scenarios with a narrow scope of visual and textual contexts. These models often fall short when faced with complex comprehension tasks, which involve navigating through a plethora of irrelevant and potentially misleading information in both text and image forms. To bridge this gap, we introduce a new, more demanding task known as Interleaved Image-Text Comprehension (IITC). This task challenges models to discern and disregard superfluous elements in both images and text to accurately answer questions and to follow intricate instructions to pinpoint the relevant image. In support of this task, we further craft a new VEGA dataset, tailored for the IITC task on scientific content, and devised a subtask, Image-Text Association (ITA), to refine image-text correlation skills. Our evaluation of four leading closed-source models, as well as various open-source models using VEGA, underscores the rigorous nature of IITC. Even the most advanced models, such as Gemini-1.5-pro and GPT4V, only achieved modest success. By employing a multi-task, multi-scale post-training strategy, we have set a robust baseline for MLLMs on the IITC task, attaining an $85.8\%$ accuracy rate in image association and a $0.508$ Rouge score. These results validate the effectiveness of our dataset in improving MLLMs capabilities for nuanced image-text comprehension.

## 1 INTRODUCTION

The swift advancement of Multi-modal Large Language Models (MLLMs) OpenAI et al. (2024); Team et al. (2024); Chen et al. (2023); Liu et al. (2023); Li et al. (2023b) has recently showcased their remarkable aptitude for tackling vision-language tasks. These models have significantly improved in areas such as logical reasoning Yue et al. (2024); Han et al. (2023), high-resolution image comprehension Dong et al. (2024), In-Context Learning (ICL) Zeng et al. (2024); Baldassini et al. (2024), and Chain of Thought (COT) processing Ge et al. (2023a), leading to notable achievements. Despite these advancements, current applications typically engage MLLMs in the basic multi-modal comprehension task, which involves posing questions tied to a narrow scope of visual content. This approach falls short when compared to human daily comprehension.

Human comprehension, particularly in document analysis, presents unique challenges not found in basic comprehension tasks such as Visual Question Answering (VQA) tasks Antol et al. (2015). As shown in Fig. 1, these challenges include: 1) **Interleaved Text-Image Distraction**: MLLMs are forced to find the correct context from irrelevant text-image mixtures. 2) **Long-Form Content**: The inputs consist of intertwined sequences of images and text, demanding comprehension over long multimodal context. 3) **Image-Referenced Answers**: The model is tasked with identifying and linking the relevant image and text based on specific instructions before crafting an appropriate response.

---

[*]Equal Contribution
[†]Corresponding Author
[♠]Project Leader

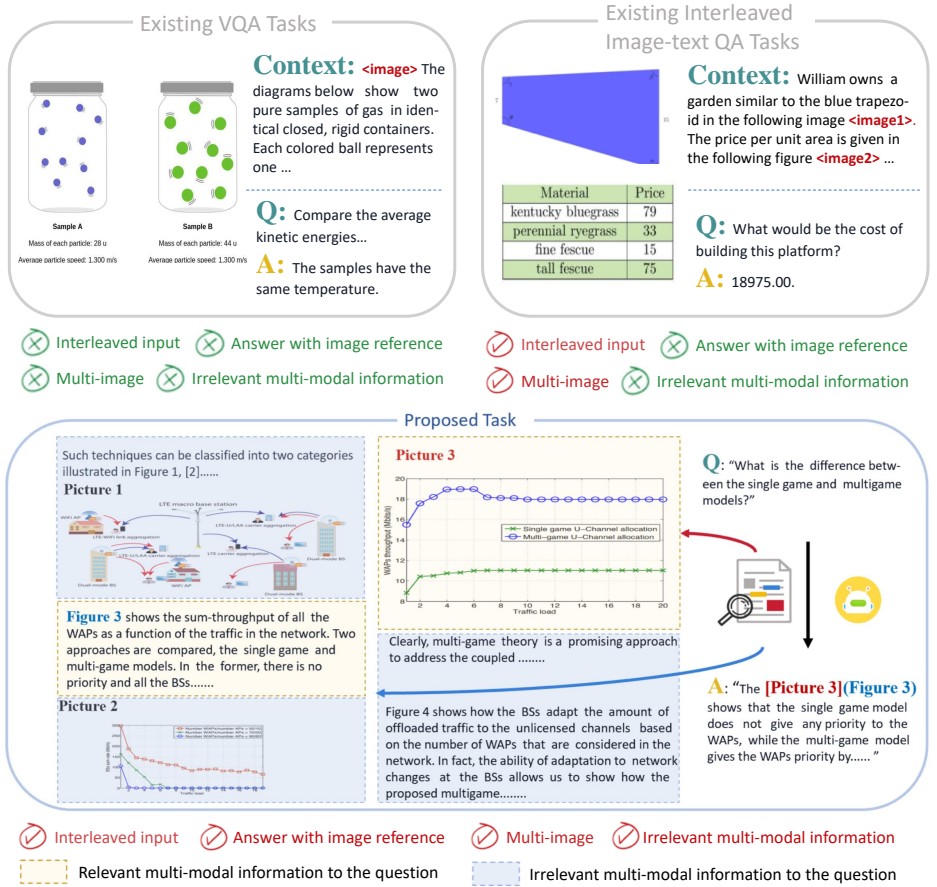

Figure 1: Comparison between existing VQA and IITC tasks. While existing VQA tasks use single images with relevant context, IITC incorporates multiple images and extended text with both relevant and irrelevant information. Models must specify reference images in their responses.

Recent prominent open-source MLLMs, such as LLaVA 1.5 Liu et al. (2023), CogVLM Wang et al. (2023b), and PaperOwl Hu et al. (2023), lack support for multiple image inputs and struggle with such intricate comprehension task. Multi-image, multi-turn models Chen et al. (2023) and video-based MLLMs Li et al. (2024) face challenges in concurrently processing extended visual and textual contexts effectively. Additionally, benchmarks for evaluating such an intricate task are limited, obscuring our understanding of MLLMs' practical comprehension abilities.

To bridge this gap, we introduce a novel multi-modal task, Interleaved Image-Text Comprehension (IITC), which involves locating the relevant text and image within a complex context given a question, providing an accurate answer, and outputting the corresponding image index. Meanwhile, we present the first benchmark MLLM to handle the above challenges within the IITC task.

Specifically, we observe that IITC is a complex task where simply increasing the number of Interleaved Image-Text QA samples is insufficient for accurate comprehension. Therefore, we propose utilizing a multi-task learning strategy to enhance the model's understanding capabilities: employing the Image-Text Association (ITA) auxiliary task to strengthen the model's ability to accurately locate image-text paragraph correspondences. In addition, we introduce a multi-scale training strategy, constructing the training data with progressively increasing context lengths and image numbers to enhance the model's robustness against gradually increasing redundant image-text content.

Ultimately, we have developed a novel dataset, designated as VEGA. It is comprised of two subsets, one tailored for the IITC task and another for the ITA task. The longest interleaved image-text content in VEGA reaches up to 8 images and 8k tokens. We design the instruction of the IITC task to be a question about only one of the images, requiring the model to specify the image it refers to in

its answer. We assess the model's interleaved image-text reading comprehension ability by both the correct rate of associated images, and the text quality of the answer by ROUGELin (2004) and BLEU Papineni et al. (2002). We have evaluated several state-of-the-art MLLMs on our dataset, validating the challenge of our tasks. Furthermore, we have fine-tuned the Qwen-VL-Chat model Bai et al. (2023) on the VEGA dataset to set a robust baseline for the IITC task.

In conclusion, this study makes several noteworthy contributions to the field:

- **Identifying a Novel Task.** We introduce a novel Interleaved Image-Text Comprehension (IITC) task. It evaluates and enhances MLLMs' capabilities of following complex instructions and extracting key cues in long and interleaved image-text scenarios.
- **Introducing the VEGA Dataset.** We develop a new VEGA Dataset for the IITC task that enables a comprehensive understanding of scientific literature, whose multi-modal context reaches up to 8,000 tokens in length and contains up to 8 images.
- **Benchmarking MLLMs in IITC.** We utilize the VEGA dataset to appraise the IITC capabilities of current state-of-the-art models, including GPT4V, Gemini-1.5-pro, and Qwen-VL-Chat, experimentally validating the challenge of IITC.
- **Enhancing MLLMs' IITC Capabilities.** We fine-tune the Qwen-VL-Chat model on the VEGA dataset with a multi-scale, multi-task training strategy and our experimental findings demonstrate that it achieves an image association accuracy rate of 85.8%. This represents a significant improvement and establishes a strong baseline for the IITC task.

## 2 RELATED WORK

### 2.1 MLLMs

Recently, there has been a notable increase in the deployment of MLLMs aimed at tackling more complex tasks that involve multiple forms of media. These advanced models are equipped to understand and interpret not just text, but also visual content like images and videos Ge et al. (2023b). For instance, BLIP-2 Li et al. (2023a) features the Q-Former, a key component that establishes a link between a static LLM and visual data, showing impressive results in VQA tasks. InstructBLIP Dai et al. (2024) is specifically fine-tuned using a variety of instruction-based datasets, which improves its understanding of visual scenes and dialogues. Multi-modal-CoT Zhang et al. (2023) brings the concept of chain-of-thought Wei et al. (2022) into the multi-modal domain, demonstrating strong performance on the ScienceQA benchmark Lu et al. (2022). LLaVA Liu et al. (2024b) uses a linear approach and fine-tunes the entire LLM to enhance its effectiveness. The LLaVA-NeXT Liu et al. (2024a), compared to LLaVA-1.5, quadruples the pixel resolution of input images, improves visual guidance for data blending, and offers enhanced visual reasoning and OCR capabilities. In this work, our focus is on the IITC task, so we concentrate only on models that support multi-image input, such as Qwen-VL Bai et al. (2023) and InternVL Chen et al. (2023). We use Qwen-VL-Chat as the base model for training to establish a baseline for the IITC task.

### 2.2 MLLM BENCHMARKS

The evolution of MLLMs, driven by multi-modal pretraining and instruction tuning, has outpaced traditional benchmarks like visual question answering and image captioning. New benchmarks have emerged to assess aspects such as OCR capabilities, adversarial robustness, and hallucination susceptibility. POPE Li et al. (2023c), HaELM HaELM Wang et al. (2023a), LAMM Yin et al. (2023), LVLM-eHub Xu et al. (2023), and MM-Vet Yu et al. (2023) provide insights into these areas and give a holistic view of MLLM's performance.

Vision-language benchmarks like Winoground Thrush et al. (2022), RAVEN Zhang et al. (2019) , OK-VQA Marino et al. (2019), and VCR Zellers et al. (2019) aim to measure MLLMs' nuanced reasoning abilities. TextVQA Wright et al. (2010), FigureQA Kahou et al. (2017), ScienceQA Lu et al. (2022), and MathVista Lu et al. (2023) contribute to understanding MLLMs' domain-specific reasoning. Comprehensive benchmarks like MME Fu et al. (2024), MMBench Fu et al. (2024), and SEED-Bench cover a variety of reasoning skills. However, these benchmarks only evaluate model performance in scenarios with limited single-image context. To address this limitation, we created the VEGA dataset.

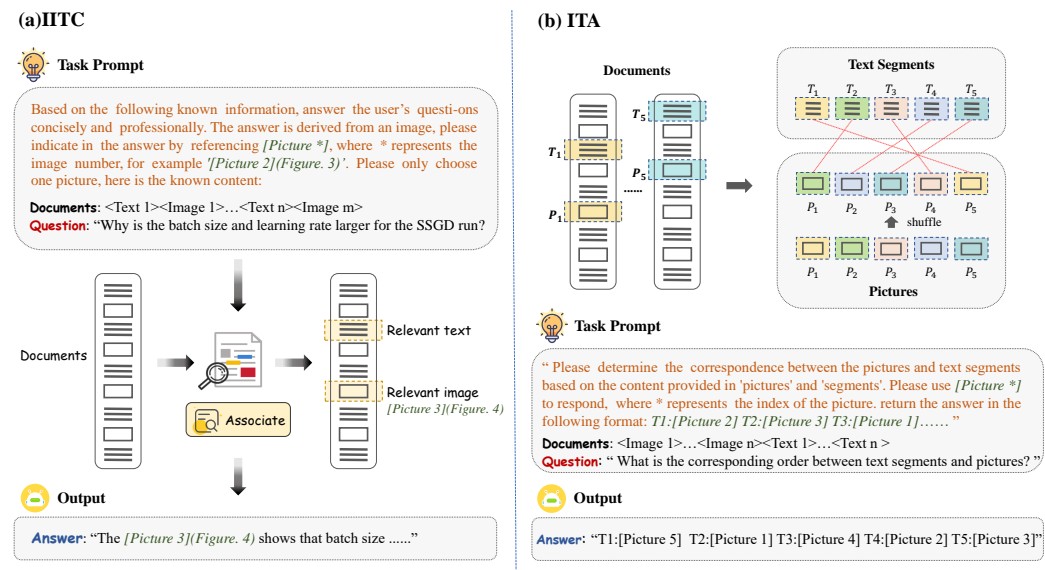

Figure 2: The task definition of IITC and ITA tasks. (a) The IITC task takes long interleaved image-text content as input and requires the model to specify the image it refers to in its response. (b) The ITA task takes shuffled images and text segments from different articles as input and requires the model to output the relationship between the text and the images. <Text *> and <Image *> represent a text segment and an image, respectively. They are both tokenized and fed into the model along with the task prompt and the question.

## 3 PROPOSED METHOD

In this section, we introduce our VEGA dataset by first elaborating the task definition (3.1), followed by outlining the process of construction (3.2) and statistics (3.3). Finally, we present the training details of our baseline MLLM model (3.4).

### 3.1 TASK DEFINITION

**Interleaved Image-Text Comprehension (IITC).** Our VEGA dataset is meticulously designed to address the challenging IITC task, with the goal of advancing the real-world comprehension capabilities of MLLMs. As demonstrated in Fig. 1, an MLLM is presented with the complex task of discerning relevant text and images in response to a given question. The model is expected to not only deliver an accurate answer but also to identify the specific image index related to the response. This requirement extends beyond the traditional boundaries of VQA tasks Antol et al. (2015).

To facilitate this process, we have developed a specialized prompt that directs MLLM to carry out the desired instruction. The complete input for MLLM is organized as shown in Fig. 2 (a). We utilize the notation "[Picture n]" within the text to establish a clear reference to the n-th image, whether it appears in various sections of an article or across different examples. This approach enables the model to learn a consistent and standardized format for image referencing.

In the IITC task, the crux of the challenge lies in the model's proficiency in accurately linking the appropriate images with the corresponding textual information as dictated by the instructions. This precise pairing is crucial, for it is the foundation upon which a coherent answer is constructed. To quantitatively gauge the model's aptitude for this association, we have crafted each question pertaining to a specific input image, necessitating that the model explicitly reference this image as part of its response.

The model's prowess in image-text association is quantified by its success rate in identifying the correct image. Meanwhile, the textual response's quality is evaluated using established linguistic

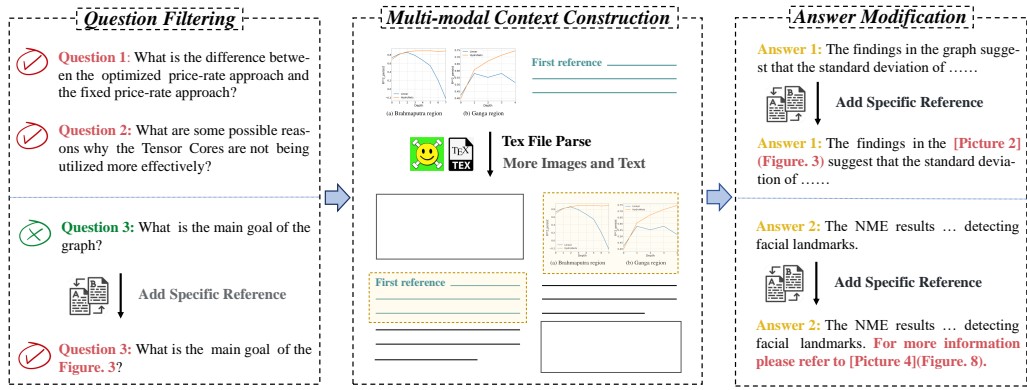

Figure 3: The construction process of the VEGA dataset. We use the SciGraphQA dataset as a foundation, to which we add more images and text to its context, modify questions and answers that lack clear image references, and incorporates references to meet the requirements of the IITC task.

metrics, namely the ROUGE Lin (2004) and BLEU Papineni et al. (2002) scores. Additionally, inspired by LLaVA-Bench Liu et al. (2023), we also evaluated the text quality using the advanced LLM Llama-3.1-70B-Instruct Dubey et al. (2024), the prompt temple we used could be found in the appendix. Together, these measures provide a comprehensive assessment of the model's performance on the IITC task.

**Image-Text Association (ITA)**. It is the auxiliary task designed to sharpen MLLM's precision in aligning text paragraphs with their relevant images. As illustrated in Fig. 2(b), ITA challenges the model with a shuffled array of inputs: $(P_1, \ldots, P_n, T_1, \ldots, T_n)$. Each image, labeled as $P_i$ from $i = 1$ to $n$, is linked to a unique article, while each text snippet $T_i$ provides context for its image $P_i$. The textual inputs are intentionally scrambled at the outset. ITA's goal is for the model to accurately match and declare the correct pairings of each text segment $T_i$ with its image $P_i$, as demonstrated in Fig. 2(b). Compared to the IITC task, ITA places greater emphasis on the model's ability to associate images with text. It is anticipated that training with ITA will bolster this skill within the model, leading to improved performance on the IITC task. For the ITA task evaluation, with a set of $n$ images, we count the number of correctly associated image-text pairs $m$, and use the ratio $m/n$ as the performance metric for the ITA task.

## 3.2 DATA COLLECTION

We develop the VEGA dataset upon the foundation of SciGraphQA Li & Tajbakhsh (2023). Sci-GraphQA is a multi-turn question-answering dataset for scientific graphs, containing 295k high-quality, multi-turn data entries. It collected part of the papers published on arXiv from 2010 to 2020, and extracted the pictures (denoted by $\{P_i\}_{i=1}^N$, $N$ is the total number of pictures), the captions ($\{C_i\}_{i=1}^N$) of the pictures, and the first paragraphs ($\{M_i\}_{i=1}^N$) mentioning the pictures from the papers.

As outlined in Fig. 3, our enhancement involves a meticulous process of questions filtering that pertain to the visual-textual context, constructing long visual and textual context, and formulating answers that directly reference images. The VEGA dataset is structured into two subsets, each specifically curated to train models on the IITC and ITA tasks, respectively.

### 3.2.1 SUBSET FOR IITC

**Question Filtering**. In the original SciGraphQA dataset, some questions about images are quite general, such as "What are the implications of the results shown in the graph?" and "What does the x-axis and y-axis of the graph represent?" For more examples, please refer to the supplementary materials. However, when these questions are placed within the context of multi-image long articles, they encounter issues with ambiguous image references. Furthermore, the answers are heavily dependent on the image content, rather than the multi-modal context information, which is not

aligned with the objectives of the IITC task. To address this, as illustrated in Fig. 3, we modify the training set by reducing the proportion of such data and incorporating explicit image references in the questions, thereby enhancing the model's image-reading capabilities. For the test set, we filter out these questions and focus on evaluating the model's ability to comprehend complex interactions between text and images based on specific queries. Through manual screening, we curate over 2,000 questions and selecte 700 high-quality questions to form the final IITC test set.

**Context Construction**. We investigate two approaches for crafting extended multi-modal contexts. The first approach involves randomly merging images and their introductory paragraphs from various papers in the original SciGraphQA dataset into a long context designated as $< P_i^k M_i^k >< P_j^l M_j^l > ... < P_n^m M_n^m >$ where $k$, $l$ and $m$ denote different papers, and then posing questions on each image and its textual surroundings. The second expands the text-image sequence from the original paper based on the paragraph $\{M_i\}_{i=1}^N$. The expanded sequence is denoted as $\{E_i\}_{i=1}^N$, which can maintain the structure of the paper and capture the inherent multi-modal context.

For the IITC task's test set, we utilize the latter approach. During the training set construction, we assess models trained with both approaches, noting superior performance with the second, as mentioned in Sec. 4.3. Beyond the advantage of consistency in data construction, we identify two key reasons for this outcome. First, the disparity in image-text content across different papers is significant, and the first approach does not effectively support the model's ability to discern between relevant and distracting multi-modal information, thereby hindering its capability to accurately match questions with the correct image-text responses. Second, the brevity of the first mention paragraph $M_i$ often results in an insufficient textual context, impeding question comprehension and response. Consequently, we select the second one for assembling the multi-modal long context for IITC.

We elaborate on the second approach of context construction. Illustrated in Fig. 3, we retrieve the paper's Tex files from Arxiv based on the 'id' field from the original SciGraphQA dataset. We then match the 'caption' field with TeX commands '\caption{.}', '\label{.}', and '\ref{.}' to pinpoint the paper's paragraph that first mentions the picture. Next, we expand this paragraph's context by sequentially incorporating lines from the adjacent text. We intersperse 2 to 8 random pictures from the document into this text until achieving a specific context length. To evaluate the model's efficacy across varying token lengths, we craft two dataset versions capped at 4k and 8k tokens, ensuring a balanced token count distribution as depicted in Fig. 4. To accurately represent the text-image spatial relationship, we maintain the sequence and placement of the inserted images as they appear in the TeX files. Lastly, we substitute the '\ref{.}' markers related to the embedded images with the notation "[Picture n]", clarifying the link between these references and their corresponding images.

**Answer Modification**. IITC requires the model to specify the image it refers to in its response. To meet this requirement, we modify the answers in the original dataset. We replace the subjects in the original answers, such as 'graph' and 'figure', with the notation [Picture $i$](Figure $j$), where $i$ denotes the sequence number of the input image, and $j$ signifies the image's index within the original document. Alternatively, we add 'For more information, please refer to [Picture $i$](Figure $j$).'

### 3.2.2 SUBSET FOR ITA

We craft the Image-Text Association (ITA) task to bolster the model's proficiency in correlating text with images and to fortify its accuracy in image indexing responses. For each image, we generate a textual context $T_i = E_i$ akin to the IITC task, extending the text around the first mention paragraph $M_i$, capped at 2,000 characters. Note that in the ITA task, the expanded context paragraphs $E_i$ does not include images outside of $P_i$.

We streamline the ITA task by pairing images with text from disparate articles, randomizing them, and prompting the model to reorder these pairs. Despite the task's complexity, leading-edge methods like Qwen-MAX Qwen-VL Team (2024) and Gemini-1.0-pro Team et al. (2024) exhibit only moderate success, particularly as the volume of pairs escalates, as evidenced in Table 2. At the same time, we find that direct training with expanded image-text pairs is insufficient for the model to master text-image associations. To mitigate the learning challenge, we implement a multi-scale training strategy in the ITA task. We design three textual scales for the training set, corresponding to the image caption $C_i$, the first mention paragraph $M_i$, and the expanded context paragraphs $E_i$. We also design two image quantity scales, with sets of three and five images. Experiments have shown that

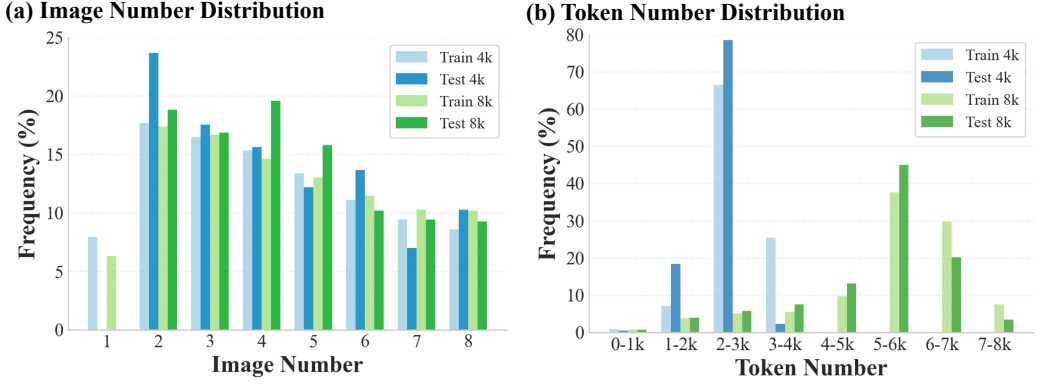

Figure 4: The distribution of the number of images and tokens in the IITC subset of the VEGA dataset. The number of tokens for each image is 256.

this multi-scale training set construction method can make the model more adaptable to the ITA task. In testing, we evaluate the model using solely the extended pairs, independent of other scaled data.

| Subset | Scale | Train | Test | Total |
|--------|-------|-------|------|-------|
| IITC | 4k Tokens | 208103 | 700 | 208803 |
| | 8k Tokens | 196947 | 700 | 197647 |
| ITA | 3 Pic C. | 23991 | - | 23991 |
| | 3 Pic M. | 23993 | - | 23993 |
| | 3 Pic E. | 47983 | 500 | 48483 |
| | 5 Pic C. | 23986 | - | 23986 |
| | 5 Pic M. | 23985 | - | 23985 |
| | 5 Pic E. | 45226 | 486 | 45712 |

Table 1: The composition of the VEGA Dataset.

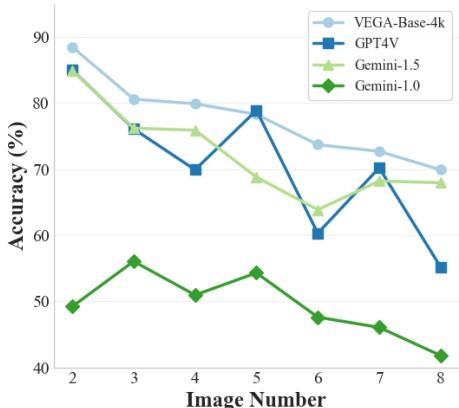

Figure 5: The relationship between accuracy and number of images in the IITC task.

### 3.3 DATASET STATISTICS

The VEGA dataset is an extensive repository featuring 50k scientific literature entries, over 200k question-and-answer pairs, and a rich trove of 400k images. It includes the IITC subset, which is segmented into two categories based on token length: one supports up to 4,000 tokens, while the other extends to 8,000 tokens. Here, images are equated to 256 tokens each. Both categories offer roughly 200k training instances and 700 meticulously curated, high-caliber test samples. The ITA task is categorized into six divisions, with two dedicated to image quantity and three to text length. Table. 1 presents the train and test data statistics for the VEGA dataset, while Fig. 4 details the distribution of image numbers and token counts within the IITC subset, providing insights into the visual and textual context lengths.

### 3.4 BASELINE MODEL

To further validate our dataset, we fine-tune the Qwen-VL-Chat Bai et al. (2023) model at two distinct maximum token lengths, 4k and 8k, training a dedicated model for each configuration, denoted

Table 2: Evaluation results for various MLLMs. Models that are not fine-tuned and have the best performances are indicated with underlines. "LB." refers to the LLaVA Bench score, "LB.T" indicates the LLaVA Bench score when the image is correct, and "LB.F" represents the LLaVA Bench score when the image is incorrect.

| Model | IITC 4k | | | | | | IITC 8k | | | | ITA | |
|---|---|---|---|---|---|---|---|---|---|---|---|---|
| | Rouge-L | BLEU | Acc | LB. | LB.T | LB.F | Rouge-L | BLEU | Acc | LB. | 3 pic | 5 pic |
| InternVL-1.5 | 0.378 | 0.128 | 0.364 | 6.37 | 6.74 | 6.16 | 0.374 | 0.126 | 0.292 | 6.19 | 0.238 | 0.014 |
| Qwen-VL-Chat | 0.323 | 0.093 | 0.002 | 4.76 | 5.21 | 4.76 | 0.250 | 0.065 | 0.001 | 3.98 | 0.043 | 0.000 |
| Qwen2-VL-Instruct | 0.431 | 0.137 | 0.017 | 6.55 | 6.92 | 6.54 | 0.393 | 0.124 | 0.006 | 6.36 | 0.385 | 0.029 |
| Qwen-MAX | 0.356 | 0.107 | 0.684 | 5.20 | 5.47 | 4.60 | 0.343 | 0.107 | 0.365 | 5.05 | 0.231 | 0.017 |
| Gemini-1.0-pro | 0.415 | 0.138 | 0.649 | 5.91 | 6.19 | 5.39 | 0.404 | 0.132 | 0.501 | 5.81 | 0.570 | 0.092 |
| Gemini-1.5-pro | 0.354 | 0.087 | 0.753 | 6.86 | 7.10 | 6.11 | 0.363 | 0.090 | 0.737 | 7.03 | 0.956 | 0.891 |
| GPT4V | 0.342 | 0.091 | 0.805 | 7.24 | 7.43 | 6.47 | 0.322 | 0.085 | 0.752 | 7.13 | 0.912 | 0.538 |
| VEGA-Base-4k | 0.508 | 0.252 | 0.858 | 5.65 | 5.87 | 4.31 | 0.488 | 0.228 | 0.789 | 5.43 | 0.998 | 0.992 |
| VEGA-Base-8k | 0.492 | 0.241 | 0.845 | 5.58 | 5.83 | 4.23 | 0.473 | 0.223 | 0.775 | 5.41 | 0.994 | 0.936 |
| Ground Truth | - | - | - | 7.67 | - | - | - | - | - | 7.78 | - | - |
| Random Guess | - | - | 0.227 | - | - | - | - | - | 0.227 | - | 0.167 | 0.008 |

as VEGA-Base-4k and VEGA-Base-8k. For more training detail, please see our supplementary materials.

# 4 EXPERIMENTS

## 4.1 EXPERIMENTAL SETUP

We evaluate the performance of the current state-of-the-art MLLMs on the IITC and ITA tasks. We test three open-source MLLMs: 1) InternVL-1.5 13BChen et al. (2023), 2) Qwen-VL-Chat 7B Bai et al. (2023), and 3) Qwen2-VL-Instruct 7B Wang et al. (2024), as well as four advanced proprietary models: 1) Gemini-1.5-pro Team et al. (2024), 2) Gemini-1.0-pro Team et al. (2024), 3) GPT4V OpenAI et al. (2024), and 4) Qwen-Max Qwen-VL Team (2024). Finally, we evaluate the performance of our baseline models VEGA-Base-4k and VEGA-Base-8k.

## 4.2 MAIN RESULTS

Table 2 presents the evaluation results of mainstream state-of-the-art MLLMs on the VEGA dataset, highlighting the significant challenges posed by the IITC and ITA tasks. InternVL, Qwen-VL-Chat, and Qwen2-VL-Instruct exhibit poor performance in the ITA task and image accuracy on the IITC task, which can be attributed to these open-source models' limited ability to follow instructions. As a result, they struggle significantly with the complex contextual inputs and directives inherent in these tasks.

Among the proprietary models, GPT4V achieves the highest image-relation accuracy on the IITC task. However, error analysis reveals that the primary issues include interference from similar images and instability in following instructions. Gemini-1.5-pro demonstrates exceptional accuracy in the ITA task, underscoring its robust image-text comprehension capabilities. In terms of text generation quality, Gemini-1.0-pro scores higher on the ROUGE-L and BLEU metrics compared to Gemini-1.5-pro and GPT4V. This discrepancy arises from two main factors: (1) Gemini-1.5-pro and GPT4V tend to summarize using their own language rather than directly incorporating descriptions from the original text, and (2) GPT4V's outputs are often longer than the Ground Truth, which disadvantages it in automated evaluations. Nonetheless, both Gemini-1.5-pro and GPT4V achieve higher LLaVA Bench scores.

We also collected the average scores of each model in the IITC 4K task under correct image selection and incorrect image selection, denoted as LB.T and LB.F, respectively. The LB.T score is much higher than the LB.F score, indicating that image-text association is crucial for the quality of model answers.

Our model, trained using Qwen-VL-Chat 7B on the VEGA dataset, achieves state-of-the-art results across all metrics in both tasks, except for the LLaVA Bench score. After training, Qwen-VL-

Table 3: Analysis of the impact of VEGA on general VQA performance. The base model we used is Qwen-VL-Chat. The VQA dataset comprises training data from VQAv2, TextVQA, and ChartQA in a 1:1:1 ratio. When training with both the VEGA and VQA datasets, the data ratio is 1:1.

| Model Name | Training Set | MMMU | MME | ChartQA | IITC 4k | | | |
|---|---|---|---|---|---|---|---|---|
| | | | | | Rouge-L | BLEU | Acc | LB. |
| Qwen-VL-Chat | - | 0.330 | 1430.40 | 48.40 | 0.323 | 0.093 | 0.002 | 4.76 |
| Qwen-VEGA | VEGA | 0.320 | 1386.05 | **64.00** | **0.498** | **0.247** | **0.836** | **5.59** |
| Qwen-VQA | VQA | **0.340** | **1472.11** | 61.12 | 0.063 | 0.001 | 0.044 | 2.21 |
| VEGA-VQA | VEGA&VQA | 0.336 | 1460.21 | 61.52 | 0.490 | 0.235 | 0.823 | 5.37 |

Table 4: Analysis on the effectiveness of multi-task and multi-scale training strategies.

| Multi-Task | Multi-Scale | IITC 4k | | | IITC 8k | | | ITA | |
|---|---|---|---|---|---|---|---|---|---|
| | | Rouge-L | BLEU | Acc | Rouge-L | BLEU | Acc | 3 pic | 5 pic |
| w/o | - | 0.501 | 0.247 | 0.838 | 0.478 | 0.220 | 0.766 | - | - |
| w | w/o | 0.502 | 0.247 | 0.833 | 0.485 | 0.228 | 0.779 | 0.155 | 0.004 |
| w | w | **0.508** | **0.252** | **0.858** | **0.488** | **0.228** | **0.789** | **0.998** | **0.992** |

Chat's LLaVA Bench score improved from 4.76 to 5.65, surpassing the closed-source model Qwen Max. This demonstrates the effectiveness of our dataset in enhancing the model's ability to process interleaved image-text inputs.

## 4.3 ABLATION STUDY

**Impact of VEGA on General VQA Performance** To investigate the impact of the VEGA dataset on the model's general VQA capabilities, we utilized VLMEvalKit Duan et al. (2024) to test the model's performance on MMMU Yue et al. (2024), MME Fu et al. (2024), and ChartQA Masry et al. (2022).

Based on the experimental results in Table 3, we draw the following conclusions:

1. Qwen-VEGA shows some decline in performance on MMMU and MME compared to Qwen-VL-Chat. This can be attributed to two main factors: 1) Qwen-VEGA focuses on enhancing the model's ability to understand long-form text-image content within the VEGA dataset (specifically in the scientific paper domain), which may negatively impact its VQA capabilities in other scenarios. The improved performance of VEGA-VQA suggests that this issue can be mitigated through joint SFT training. 2) After training on the VEGA dataset, Qwen-VEGA tends to produce longer outputs. Although VLMEvalKit has powerful post-processing code for answers, there are still cases where the scores are lower. (In the MME test, the average output length is Qwen-VEGA: 102.32; Qwen-VL-Chat: 3.00; VEGA-VQA: 2.57).

2. Qwen-VEGA achieves high scores on ChartQA, likely due to the large number of chart-containing images in the VEGA dataset. We believe that training on the VEGA dataset enhances the model's ability to interpret chart images.

3. The similar scores of VEGA-VQA and Qwen-VQA indicate that incorporating the VEGA dataset into training does not compromise the model's VQA capabilities.

In summary, we recommend using the VEGA dataset for multi-task joint training with other multi-modal datasets. This approach can enhance the model's ability to handle mixed text-image inputs while preserving its traditional VQA capabilities, making it more adaptable to a broader range of instructions and improving its generalization abilities.

**Multi-task and Multi-scale Learning.** We investigate the impact of multi-task and multi-scale training strategies on model training. As shown in Table 4, our base model is Qwen-VL-Chat, where "Multi-Task" represents joint training on both the IITC and ITA tasks, while "without Multi-Task" indicates training solely on the IITC task. "Multi-Scale" denotes the use of a multi-scale training strategy in the ITA task, otherwise training is only conducted on the extended context scale of the ITA task. The experimental results show that the model employing both Multi-Scale and Multi-Task

Table 5: Performance comparison of the base model on SciGraphQA and IITC 8k Task with different training sets. VEGA* denotes the first context construction method for the IITC task that integrates images and their initial mentioning paragraphs from several papers, as detailed in Section 3.2.1.

| Base model | Training Set | SciGraphQA | | IITC 8k | | |
|---|---|---|---|---|---|---|
| | | Rouge-L | BLEU | Rouge-L | BLEU | Acc |
| Qwen-VL-Chat | SciGraphQA | **0.538** | **0.290** | 0.406 | 0.153 | 0.000 |
| Qwen-VL-Chat | VEGA | 0.522 | 0.266 | **0.473** | **0.223** | **0.775** |
| Qwen-VL-Chat | VEGA* | 0.507 | 0.241 | 0.415 | 0.160 | 0.521 |

strategies improves the accuracy of image-text association by 2.3% compared to the model without these strategies, confirming the effectiveness of the multi-scale and multi-task training strategies. We also find that without multi-scale training, the model shows almost no performance improvement on the ITA task, with accuracy comparable to random guessing. Observing the model's failure case, it organizes the pictures and text chaotically. Such failure originates from two primary causes: First, the substantial volume of both text and images adds complexity to the task, challenging the model's ability to grasp the connections between textual and visual contexts. This complexity hinders the model's compliance with the directives for producing the correct output. Second, the complexity of ITA is attributed to the necessity of navigating multiple visual-textual relationships rather than identifying a single image in the answer. By adopting a multi-scale approach that streamlines the learning process during training, we have improved the model's compliance with instructions, leading to a significant enhancement in its performance on the ITA task. Additionally, we evaluate the impact of different lengths of visual context on the model performance in Fig. 5.

**Comparison of SciGraphQA and VEGA.** We trained Qwen-VL-Chat using the SciGraphQA Li & Tajbakhsh (2023) and VEGA datasets and tested the models' performance on these two datasets. As shown in Table 5, the model trained with the VEGA dataset performed better on both tasks, while the model trained with SciGraphQA struggled with the IITC task. These results suggest that training with the VEGA dataset not only enhanced the model's ability to handle long interleaved image-text inputs but also maintained its capability to process traditional VQA input patterns.

**Comparison of Context Construction Methods.** We assessed the impact of two context construction techniques on model performance for the IITC task. VEGA* represents the first method that builds multi-modal context for the IITC task by integrating images and their initial mention paragraphs from several papers, as detailed in Section 3.2.1. As shown in Table 5, the data reveals that VEGA* underperforms on both SciGraphQA and IITC tasks compared to our baseline model, underscoring our method's superiority. This inferior performance may stem from the fact that combining text and image context from the same paper could introduce extraneous material, inadvertently sharpening the model's proficiency in discerning relevant information from noise during training. The creation of the VEGA dataset goes beyond mere data accumulation; We carefully selected questions and constructed contexts based on the challenges faced in reading comprehension applications. This thoughtful design leads to a more effective enhancement of the model's ability to associate images and text, as well as its reading comprehension ability.

## 5    CONCLUSION AND LIMITATION

In this study, we present the VEGA dataset, tailored to boost MLLMs' Interleaved Image-Text Comprehension (IITC) in real-world applications. We employ the VEGA test set to evaluate leading MLLMs, exposing the stringent requirements of IITC tasks, particularly in multi-modal comprehension with a focus on instruction adherence. Our cutting-edge approach integrates multi-task and multi-scale learning, utilizing VEGA within the Qwen-VL-Chat framework to refine MLLMs' ability to interleave image-text comprehension and accurately generate image indexes following instructions. This approach outstrips premier proprietary models like Gemini 1.5 Pro and GPT4V, proving VEGA's potential to substantially improve IITC performance. Future enhancements to the dataset will involve embedding subtler instructions, increased image-text interactivity in answers, a wider array of data sources, and more cutting-edge evaluation methods for the IITC task.

## ACKNOWLEDGEMENTS

This work was supported by the National Science Fund for Distinguished Young Scholars (No.62025603), the National Natural Science Foundation of China (No. U21B2037, No. U22B2051, No. U23A20383, No. 62176222, No. 62176223, No. 62176226, No. 62072386, No. 62072387, No. 62072389, No. 62002305 and No. 62272401), and the Natural Science Foundation of Fujian Province of China (No. 2021J06003, No.2022J06001).

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

APPENDIX

## A TRAINING DATA

**Data Composition** During the training of the VEGA-Base model, we utilized the entire IITC subset of the VEGA dataset. As for the ITA subset, we incorporated all of the Expanded Context data, as well as half of the data from both the Captions and First Mentions. The training data distribution is visualized in Figure 6.

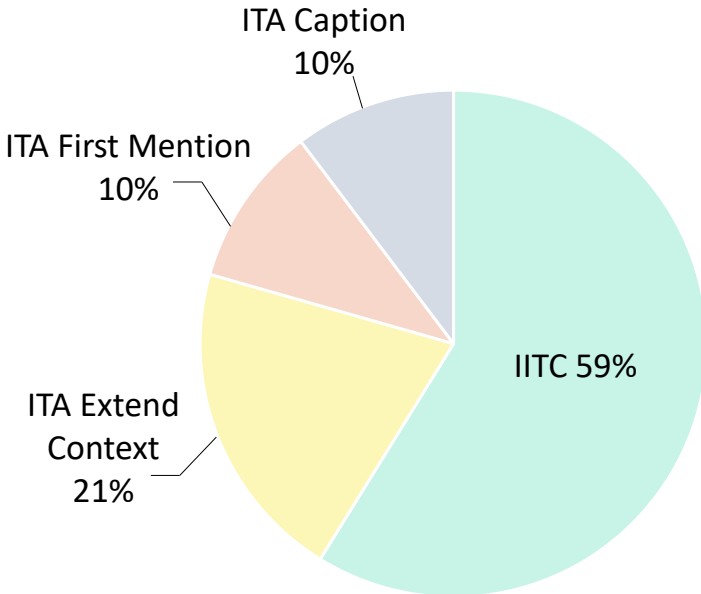

Figure 6: The composition of data for model training.

## B LLaVA BENCH PROMPT

[Picture caption]: {Picture Caption}

[Picture Context]: {Picture First Mention}

[Question]: {Question}

[Assistant 1]

{Answer 1}

[End of Assistant 1]

[Assistant 2]

{Ground Truth}

[End of Assistant 2]

[System]

We would like to request your feedback on the performance of two AI assistants in response to the user question displayed above. The user asks the question on observing an image. For your reference, the visual content in the image is represented with a few sentences describing the image.

Please rate the accuracy of their responses. Each assistant receives an overall score on a scale of 1 to 10, where a higher score indicates better overall performance.

Please only output a single line containing only two values indicating the scores for Assistant 1 and 2, respectively. The two scores are separated by a space.

Please avoid any potential bias and ensure that the order in which the responses were presented does not affect your judgment.

## C  QUESTION FILTERING

In the construction of the VEGA dataset, we filtered out the questions that lacked clear image references, as shown below:

```
What are the implications of the results shown in the graph?
What are some of the potential applications of the graph?
What are some of the implications of this graph?
Are there any other interesting aspects of the graph that you would like to highlight
What is the significance of the results shown in the graph?
What is the main goal of the experiment illustrated in the graph?
What are the key takeaways from this graph?
What are some of the key observations that can be made from the graph?
What are the key features of the mutation plot?
What are the implications of the findings in the graph?
What are the limitations of this graph?
What are the implications of the results of this graph?
Are there any limitations to the conclusions that can be drawn from the graph?
What are some of the key takeaways from this graph?
What are some of the limitations of the graph?
What are the implications of the results of the graph?
What are some of the implications of the results in the graph?
What is the significance of the different colors in the graph?
What are the limitations of the graph?
What are the implications of the results shown in the figure?
What is the main focus of the graph?
What is the significance of the 3D plot?
What is the main message of the graph?
What does the y-axis represent?
What are the key takeaways from the graph?
What can be inferred from the graph?
What is the purpose of this graph?
How does the graph support the claims made in the paper?
What are the key features of the graph?
What is the main idea of the graph?
What are the key takeaways from the figure?
What is the significance of the graph as a whole?
What is the purpose of the graph?
Are there any other interesting aspects of the graph that you would like to point out
What is the main purpose of the graph?
What is the main goal of the graph?
What is the significance of the markers in the graph?
What are the implications of this graph for future research?
What does the graph show about the performance of the proposed method?
What is the main objective of this graph?
What is the main idea of the figure?
What is the significance of the graph in the context of the paper?
What does the x-axis represent?
What is the difference between the two figures in the graph?
What does the x-axis and y-axis of the graph represent?
What are the two main lines in the graph?
What are the implications of the results in this graph?
What does the graph show?
```

```
What is the significance of this graph?
What is the main takeaway from the graph?
What are the implications of the findings from this graph?
What are the main takeaways from the graph?
What is the purpose of the two figures in the graph?
What are the key findings of the graph?
What are the two main axes of the graph?
What do the colors in the graph represent?
What are some of the implications of the findings presented in the graph?
```

## D  CASE STUDY

This section presents the performance of VEGA-Base-4k, Gemini-1.5-pro, and GPT4V on the IITC 4k task. Fig. 7 and 8 illustrates cases where all three models failed, highlighting the challenging and complex nature of the IITC task. The models were misled by deceptive textual content, resulting in incorrect answers. Fig. 9 and 10 shows instances where all three models provided the correct response. Fig. 11, 12, 13 and 14 depict scenarios where VEGA-Base-4K answered correctly, while Gemini-1.5-pro and GPT4V did not. VEGA-Base-4k successfully navigated through distractions from similar images and irrelevant text, accurately interpreting the information in the images to arrive at the correct answers, demonstrating its robust capability to process interleaved image-text inputs.

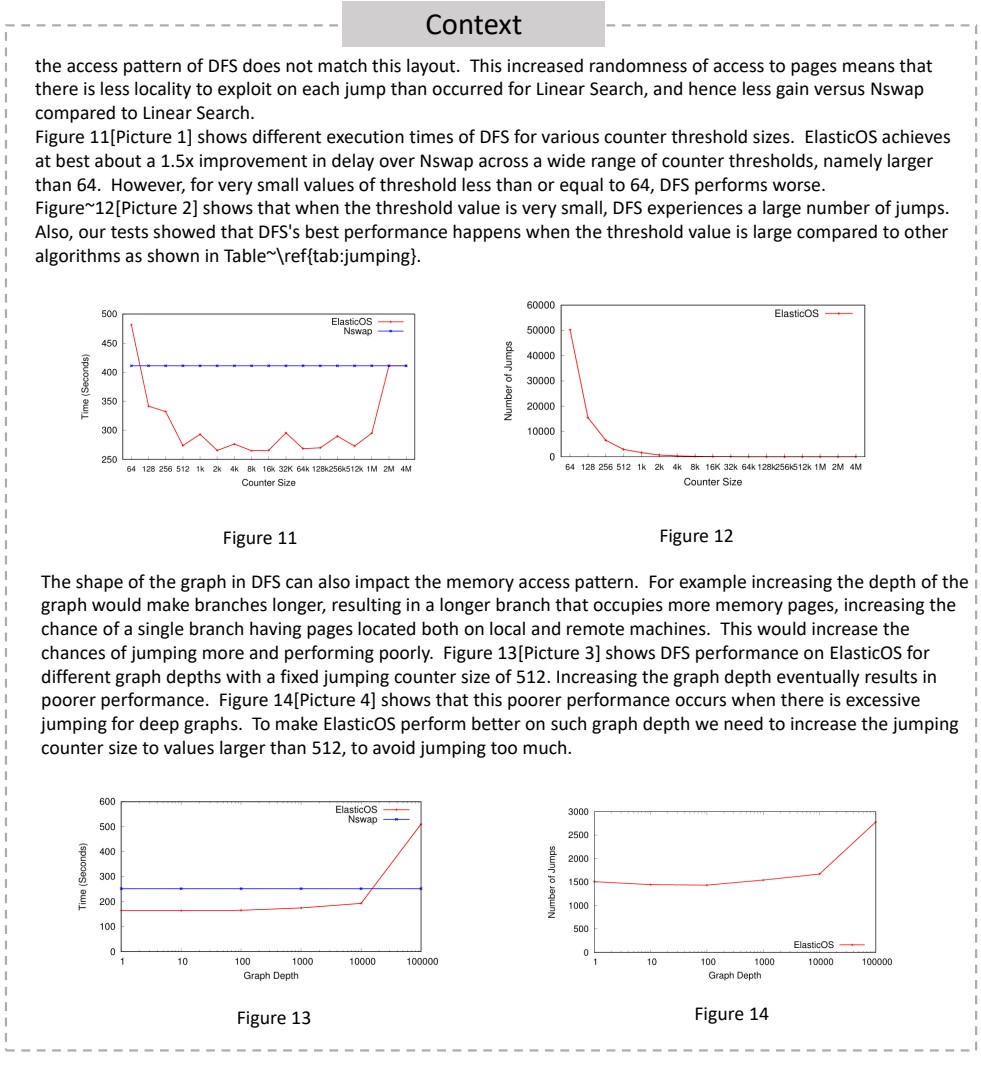

Figure 7: The context for Case 1

**Question:** What is the relationship between graph depth and performance of ElasticOS?
**Ground Truth:** The *[Picture 3](Figure. 13)* depth of a DFS search tree is the maximum number of levels in the tree. As the graph depth increases, the number of branches in the tree also increases. This can lead to longer branches, which occupy more memory pages. If a single branch has pages located both on local and remote machines, this can increase the chances of jumping more and performing poorly.

**VEGA-Base-4k:** The **[Picture 4](Figure. 14)** shows that as the graph depth increases, the performance of ElasticOS decreases. This is because as the graph depth increases, the number of jumps increases, which makes it more difficult for ElasticOS to find the shortest path.
**Score:** rouge-l: 0.260; BlEU: 0. 099

**Gemoni-1.5-pro:** The performance of ElasticOS degrades as the graph depth increases. **[Picture 3]** This is because deeper graphs lead to more jumps, as shown in **[Picture 4]**, which negatively impacts performance.
**Score:** rouge-l: 0.320; BlEU: 0.149

**GPT4V:** The relationship between graph depth and the performance of ElasticOS is that as the graph depth increases, the performance of ElasticOS tends to decrease. This is evidenced by **[Picture 3](Figure. 13),** which shows that a deeper graph results in longer execution times for ElasticOS. The performance decline is further intensified due to the increased frequency of jumping to non-local memory pages as the depth grows, which is observed in **[Picture 4](Figure. 14),** where an excessive number of jumps occur for deeper graphs. To mitigate this issue and improve performance for deep graphs, it would be necessary to increase the threshold value for the jumping counter size to more than 512.
**Score:** rouge-l: 0.349; BlEU:0.143

**Analysis:** VEGA-Base-4k may have selected the wrong image due to being misled by the related text description of [Picture 14]. In fact, [Picture 14] describes the relationship between Graph Depth and the number of jumps, indicating that VEGA-Base-4k's attention to the content of the picture might be less than that to the text content. Both Gemini-1.5 and GPT4V correctly understood the content that the context wanted to express, but they did not follow the instructions well and interpreted two pictures at the same time.

Figure 8: Models' answer and analysis for Case 1. The pictures referred in the Ground Truth are marked in blue, the incorrect images referred to in the model's answer are marked in red, and the correct images are marked in green.

## Context

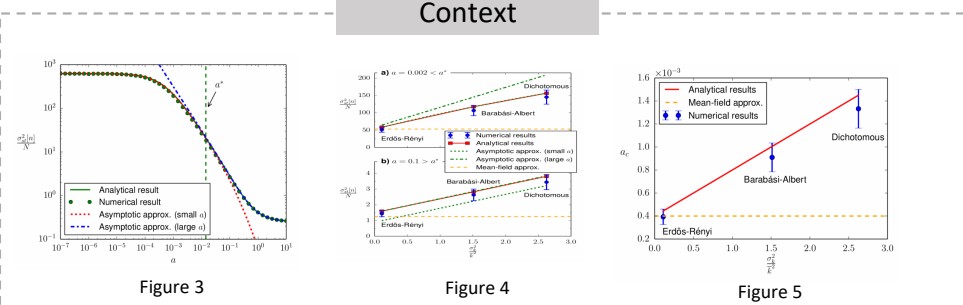

Figure 3     Figure 4     Figure 5

Let us start the description of our results by emphasizing that individual realizations of the interface density $\rho$ remain always active for any non-zero value of the noise, as it was also the case for the variable $n$ (see Fig.~\ref{f:magnetizationSingleRun}). As an example, we show in Fig.~6[Picture 4] two realizations of the dynamics for a Barab\'asi-Albert scale-free network corresponding, respectively, to the bimodal [panel \textbf{a)}] and the unimodal regime [panel \textbf{b)}]. While in the first of them ($a < a\_c$) the system fluctuates near full order, with sporadic excursions of different duration and amplitude towards disorder; in the second ($a > a\_c$), the system fluctuates around a high level of disorder, with some large excursions towards full order.

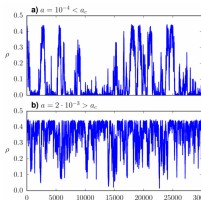

Figure 6

Introducing the annealed approximation for uncorrelated networks described above into the definition of the order parameter given in equation~\eqref{e:rho_definition}, and focusing on the steady state average value, we obtainIn this way, an explicit solution for the steady state average interface density can be found by expressing it in terms of the analytical results presented so far, namely, in terms of the variance $\sigma^2_{st}[n]$ (see Appendix ~\ref{a:order_parameter_the_interface_density} for details),This expression can be contrasted with numerical results in Fig.~7[Picture 5], where we present the steady state average interface density $\langle \rho \rangle_{st}$ as a function of the noise parameter $a$ for different types of networks. The mean-field pair-approximation result derived in \cite{Diakonova2015} is also included for comparison.

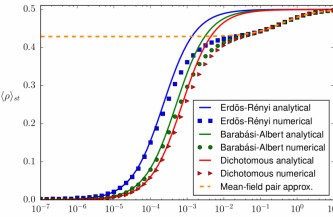

Figure 7

As we can observe in Fig.~7[Picture 5], our approach correctly captures the behavior of the system for both small ($a \lesssim 10^{-3}$) and very large values ($a \gtrsim 3$) of the noise parameter: both the asymptotic convergence towards \mbox{$\langle \rho \rangle_{st} = 0$} for small $a$ (voter model result for finite systems) and the convergence towards \mbox{$\langle \rho \rangle_{st} = 1/2$} for large $a$ (full disorder) are well reproduced. On the contrary, our analytical approach fails to reproduce the numerical results for intermediate values of the noise parameter (\mbox{$10^{-3} \lesssim a \lesssim 3$}). The origin of this discrepancy lies in the annealed network approximation: when replacing the original network by a weighted fully-connected topology, all track of local effects is lost ---precisely those measured by the order parameter. The fact that this discrepancy is only present for intermediate values of $a$ can be explained, on the one hand, by the lack of locally ordered structures in the fully disordered, large $a$ regime and, on the other hand, by the development of a global order ---more and more independent of local effects--- for decreasing values of $a$. Thus, an accurate fit of the numerical results for any value of $a$ can only be expected for topologies where local effects are absent or negligible. In the Supplementary Figure S1 presented in Appendix~\ref{a:suppementary_Figure_S1} we show that the approximation successfully fits the results in a fully-connected network. The good accuracy of the results presented above for the variance of $n$ suggests that the discrepancy between analytical and numerical results appears only when the annealed network approximation is used to derive a relationship between $\langle \rho \rangle_{st}$ and $\sigma^2_{st}[n]$, and not in the derivation of the latter, for which only global correlations are relevant.

Figure 9: The context for Case 2

**Question:** What are the three different types of networks that are studied in this figure?
**Ground Truth:** The three different types of networks that are studied in this *[Picture 5](Figure. 7)* are Erdös-Rényi random networks, Barabási-Albert scale-free networks, and dichotomous networks. Erdös-Rényi random networks are networks in which the nodes are connected randomly. Barabási-Albert scale-free networks are networks in which the nodes are connected in a preferential attachment manner. Dichotomous networks are networks in which the nodes are either connected or not connected.

**VEGA-Base-4k:** The three different types of networks that are studied in this **[Picture 5](Figure. 7)** are Erdős-Rényi, Barabási-Albert, and Dichtotomous. These networks are all random graphs, but they have different degree distributions. The Erdős-Rényi network is a random graph with a power-law degree distribution, the Barabási-Albert network is a random graph with a scale-free degree distribution, and the Dichtotomous network is a random graph with a power-law degree distribution.
**Score:** rouge-l: 0.451; BlEU: 0. 171

**Gemoni-1.5-pro:** **[Picture 5]** The three types of networks studied in this figure are: Erdős-Rényi, Barabási-Albert, and Dichotomous.
**Score:** rouge-l: 0.408; BlEU: 0.0126

**GPT4V:** The three different types of networks studied in [Picture 5](Figure. 7) are Erdős-Rényi, Barabási-Albert, and Dichotomous networks.
**Score:** rouge-l: 0.417; BlEU:0.022

**Analysis:** The text does not explicitly detail the three different types of networks depicted in [Picture 5] (Figure 7). However, the three models are correctly linked to the appropriate images and provide accurate responses. This demonstrates the model's capability for text-image correlation and image comprehension.

Figure 10: Models' answer and analysis for Case 2. The pictures referred in the Ground Truth are marked in blue, the incorrect images referred to in the model's answer are marked in red, and the correct images are marked in green.



**Context**



The experimental results show that the {\algA} factorization is verystable. **Figure 4[Picture 1] displays the growth factor of {\algA}for random matrices of size varying from 1024 to 8192 and for sizesofthe panel varying from 8 to 128.** We observe that the smaller the sizeof the panel is, the bigger the element growth is. In fact, for asmaller size of the panel, the number of panels and the number ofupdates on the trailing matrix is bigger, and this leads to a largergrowth factor. But for all panel sizes, the growth factor of {\algA} issmaller than the growth factor of GEPP. For example, for a randommatrix of size 4096 and a panel of size 64, the growth factor is onlyabout 19, which is smaller than the growth factor obtained by GEPP,and as expected, much smaller than the theoretical upper bound of$(1.078)^{4095}$

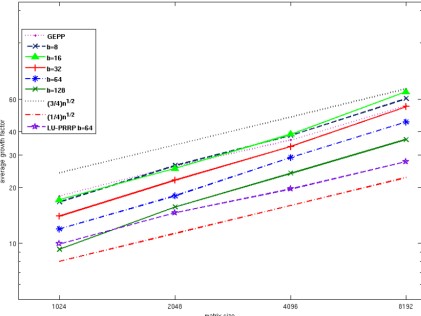

Figure 4

Tables \ref{TabRRQRLU_randn1lu} and \ref{TabLUPRRP_spec1} in Appendix Bpresent more detailed results showing the stability of the {\algA}factorization for random matrices and a set of special matrices.There, we include different metrics, such as the norm of the factors,the value of their maximum element and the backward error of the LUfactorization.We evaluate the normwise backward stability by computing three accuracy tests as performed in the HPL (High-Performance Linpack) benchmark.In HPL, the method is considered to be accurate if the values of thethree quantities are smaller than $16$. More generally, the valuesshould be of order $O(1)$. For the {\algA} factorization HPL1 is at most $8.09$,HPL2 is at most $8.04\times 10^{-2}$ and HPL3 is at most $1.60\times 10^{-2}$. We also display the normwise backwarderror, using the 1-norm,and the componentwise backward errorwhere the computed residual is $r=b-A {x}$. For our tests residualsare computed with double-working precision.

**Figure 5[Picture 2] summarizes all our stability results for{\algA}.** This figure displays the ratio of the maximum between thebackward error and machine epsilon of {\algA} versus GEPP. Thebackward error is measured using three metrics, the relative error $\|PA-LU \| / \|A \|$, the normwise backward error $\eta$, and thecomponent-wise backward error $w$ of {\algA} versus GEPP, and themachine epsilon. We take the maximum of the computed error with epsilon since smaller values are mostly roundoff error, and so taking ratios can lead to extreme values with little reliability. Results for all the matrices in our test set arepresented, that is $20$ random matrices for which results arepresented in Table \ref{TabRRQRLU_randn1lu}, and $37$ special matricesfor which results are presented in Tables \ref{TabGEPP_spec1} andalmost all ratios are between $0.5$ and $2$. For special matrices,there are few outliers, up to $ 23.71$ (GEPP is more stable) for thebackward error ratio of the special matrix \textit{hadamard} and downto $2.12\times 10^{-2}$ ({\algA} is more stable) for the backwarderror ratio of the special matrix \textit{moler}.

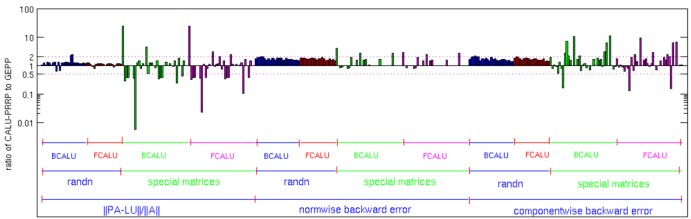

Figure 5

Between max({\algA}'s backward error, machine epsilon) and max(GEPP's backward error, machine epsilon) for all the test matrices in our set. Each vertical bar represents such a ratio for one test matrix. Bars above $10^0=1$ mean that {\algA}'s backward error is larger, and bars below $1$ mean that GEPP's backward error is larger. For each matrix and algorithm, the backward error is measured 3 ways. For the first third of the bars, labeled $\|PA - LU\| / \|A\|$, the metric is the backward error, using the Frobenius norm. For the middle third of the bars, labeled ``normwise backward error'', the metric is $\eta$ in equation \eqref{eq:eta}. For the last third of the bars, labeled ``componentwise backward error'', the metric is $w$ in equation \eqref{eq:comp}. The test matrices are further labeled either as ``randn'', which are randomly generated, or ``special'', listed in Table \ref{Tab_allTestMatrix}.

Figure 11: The context for Case 3

**Question:** What does the graph show about the stability of CALU PRRP?

**Ground Truth:** The *[Picture 2](Figure. 5)* shows that CALU PRRP is a stable algorithm, with backward errors that are at most 2.4 times larger than those of GEPP for random matrices, and at most 2 times larger for special matrices. This is consistent with the results reported in the paper, which show that CALU PRRP has a relative error of at most $9.14 \times 10^{-14}$, a normwise backward error of at most $1.37 \times 10^{-14}$, and a componentwise backward error of at most $1.14 \times 10^{-8}$ for Demmel matrix.

**VEGA-Base-4k:** The *[Picture 2](Figure. 5)* shows that CALU PRRP is more stable than GEPP for all test matrices in the set. This is evident from the fact that the backward error ratio of CALU PRRP is generally lower than that of GEPP. For example, for the special matrix hadamard, the backward error ratio of CALU PRRP is $2.12 \times 10^{-2}$, while the backward error ratio of GEPP is 23.71. This means that CALU PRRP is more accurate than GEPP for this matrix.
**Score:** rouge-l: 0.408; BlEU: 0.093

**Gemoni-1.5-pro:** [Picture 1] This graph shows the stability of the CALU PRRP algorithm by comparing its growth factor to the growth factor of GEPP for different matrix sizes and panel sizes. The growth factor of CALU PRRP is smaller than the growth factor of GEPP for all panel sizes. As the size of the panel increases, the growth factor of CALU PRRP decreases. This indicates that CALU PRRP is a more stable algorithm than GEPP, especially for larger panel sizes.
**Score:** rouge-l: 0.379; BlEU: 0.078

**GPT4V:** The graph in **[Picture 1](Figure. 4)** shows the average growth factor when using the CALU PRRP factorization for random matrices of various sizes and panel sizes. The results indicate that the smaller the size of the panel, the larger the growth factor is, which means a less stable factorization. However, for all panel sizes, the growth factor of CALU PRRP is smaller than that of GEPP (another algorithm for matrix factorization), suggesting that CALU PRRP is comparatively more stable than GEPP. The graph also indicates that, as the matrix size increases, the growth factor for CALU PRRP increases in a controlled manner, significantly below the theoretical upper bounds, showing good stability characteristics of the CALU PRRP algorithm.
**Score:** rouge-l: 0.350; BlEU: 0.0398

**Analysis:** VEGA-Base-4k picked the correct image and drew the right conclusion. Both Gemini-1.5-pro and GPT4V chose the wrong images, yet they arrived at the correct conclusions based on the text information. GPT4V's response was significantly longer compared to the Ground Truth, resulting in lower Rouge-l and BLEU scores.

Figure 12: Models' answer and analysis for Case 3. The pictures referred in the Ground Truth are marked in blue, the incorrect images referred to in the model's answer are marked in red, and the correct images are marked in green.

> ## Context
>
> For the discriminative classifiers, we train three-layer $784 \times 100 \times 10$ MLP neural network with one input layer ($784$ neurons), one hidden layer ($100$ neurons) and one output layer ($10$ neurons) for classification. In training stage, we set the learning rate as $0.01$, and the number of iterations as $5000$. SVM classifiers are trained with radial basis function (RBF) kernel for these two data sets. We built $10$ ``one-versus-all'' SVM classifiers for the $10$-class classification problem, and assign the class label with the greatest margin to the test image. The DBN used in our experiments is a $768 \times 500 \times 500 \times 2000 \times 10$ network which is trained by two stages: pre-training stage and fine-tuning stage. In its pre-training stage, each hidden layer is trained with $50$ epochs using all training data, and in its fine-tuning stage, a back-propagation training algorithm with $200$ epochs is adopted for the discriminative purpose. We refer to the interested readers to \cite{hinton2006fast} for the detailed descriptions of this deep belief network.
>
> 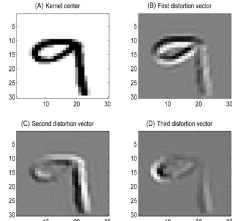
>
> Figure 2
>
> For our proposed Kernel-Distortion classifier, we use $K=100$ kernels, and set $p = 3$, $q = 40$, $\sigma_o^2 = 0.03$ and $\sigma_d^2 = 0.9$. To select better kernels, the proposed iterative kernel selection algorithm is applied. In Fig. 2[Picture 1], we show the total likelihood $Q$ as a function of iteration for up to $3000$ iterations for the digit ``$6$'', when the MNIST data set is used as an example. Although not purely monotonic, it shows a steadily increasing likelihood \footnote{A video demonstration for the process of kernel selection is provided as Supplementary Material. One can see that the class-wise likelihood is steadily increased with the number of iteration, and that the ``best to remove'' digits seem really bad at first, and then at the end, they look almost the same as the ``best to add''. }. The steady increasing likelihood demonstrates the effectiveness of our proposed iterative kernel selection algorithm. We stop the kernel-selection algorithm at the $500$-th iteration for both MNIST and USPS data sets, although the likelihood still increases. Table \ref{tab_all} shows the comparison results in terms of overall testing classification error rate. It can be shown that the discriminative classifiers usually outperform the generative classifiers for these two data sets. However, our proposed generative classifier can greatly improve the prediction performance compared with other generative classifiers and outperform some other well-trained discriminative classifiers. It performs best for USPS handwritten digits classification and ranks third for MNIST handwritten digits classification, which demonstrate the effectiveness of our proposed Kernel-Distortion classifier. LDA, GMM, GKDE, LKDE, MLP-NN, DBN, RBF-SVM, $k$-NN ($k=1$), and our Kernel-Distortion classifier. For each dataset, we highlight the best result with \underline{\textbf{Bold}} value, the second one with \textbf{Bold} value, and the third one with \textit{Italic} value, among all 11 classifiers.
>
> 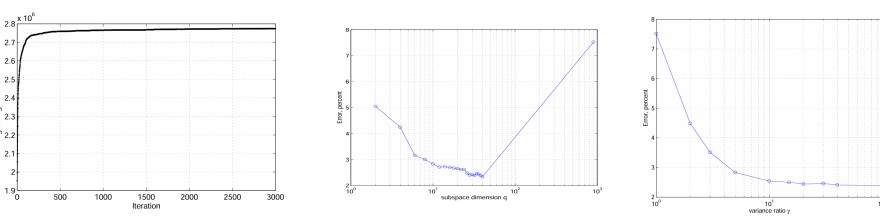
>
> Figure 3     Figure 4     Figure 5
>
> Figure. 3 The largest value of $q$ tried was 40. The performance at the far right is the Euclidean distance metric classifier, which is equivalent to $q=N^2$. Figure. 4 variance ratio $\sigma^2_d/\sigma^2_o$. Figure. 5 The parameters of the proposed approach include $p$, $q$, $K$, $\sigma_o^2$, and $\sigma_d^2$. Given the MNIST data set as an example, we further study the classification performance with different parameter values. Firstly, the performance for different $q$ values from $2$ to $40$ is measured, as shown in Fig. 3[Picture 2], when $K = 100$, $p = 3$, $\sigma_o^2 = 0.03$ and $\sigma_d^2 = 0.9$. At the far right is the performance of the Euclidean distance metric, which is equivalent to $q=N^2$. The performance is the best at $q=40$, the highest $q$ value tried in our experiments. Due to the slow performance improvement and the high storage requirement, we do not use a higher value. For $p=3$ and $q=40$, the performance as a function of $\gamma$ is shown in Fig. 4[Picture 3], indicating that $\gamma=30$ is a reasonable value. Notice that our approach is the Euclidean distance metric classifier when $\gamma=1$. We also evaluate the performance as a function of $p$. The result is shown in Fig. 5[Picture 4], which shows no advantage for $p$ larger than $3$. We note that it is still possible to achieve better parameter settings, and different classification tasks would have different optimal parameter settings.

Figure 13: The context for Case 4

**Question:** What does the graph show about the relationship between variance ratio and total classification error?

**Ground Truth:** The *[Picture 4](Figure. 5)* shows that there is a negative correlation between variance ratio and total classification error. This means that as the variance ratio increases, the total classification error decreases. This is likely because a higher variance ratio means that the data is more spread out, which makes it easier to distinguish between different classes.

**VEGA-Base-4k:** The *[Picture 4](Figure. 5)* shows that the total classification error decreases as the variance ratio increases. This is because a higher variance ratio means that the model is more sensitive to the input data, and thus is able to make more accurate predictions. However, the decrease in error is not linear, and the error eventually plateaus. This is because the model is eventually able to learn the underlying structure of the data and make accurate predictions without the need for more variance.

**Score:** rouge-l: 0.447; BlEU: 0.168

**Gemoni-1.5-pro:** The graph in **[Picture 3]** shows the relationship between the variance ratio and the total classification error. As the variance ratio increases, the error rate initially decreases, reaching its lowest point around a variance ratio of 30. After this point, the error rate remains relatively stable with only minor fluctuations as the variance ratio continues to increase.

**Score:** rouge-l: 0.349; BlEU: 0.156

**GPT4V:** The graph in **[Picture 3](Figure. 4)** shows that as the variance ratio (γ) increases, the total classification error decreases, reaching an optimal point at γ=30. Beyond this point, the error rate levels off, indicating that increasing the variance ratio further does not significantly affect the error rate. This suggests that a variance ratio of around 30 is a reasonable value for the classification task in this context.

**Score:** rouge-l: 0.349; BlEU:0.143

**Analysis:** VEGA-Base-4k picked the correct image and drew the right conclusion. Both Gemini-1.5 and GPT4V selected the wrong images and responded based on the related text content of the incorrect images, resulting in lower ROUGE scores. A possible reason could be that they were misled by the labels on the y-axis of [Picture 3](Figure. 4).

Figure 14: Models' answer and analysis for Case 4. The pictures referred in the Ground Truth are marked in blue, the incorrect images referred to in the model's answer are marked in red, and the correct images are marked in green.

