# OpenReview forum: "Learning Interleaved Image-Text Comprehension in Vision-Language Large Models"
_ICLR.cc/2025/Conference — ICLR 2025 Poster_

### Official Review · Reviewer_qCo5 · 2024-10-31

**Soundness:** 3
**Presentation:** 2
**Contribution:** 2
**Rating:** 6
**Confidence:** 4

**Summary:**

The paper introduces VEGA, a dataset designed for interleaved image and text understanding. It also proposes a multi-modal large language model (MLLM) baseline fine-tuned on this dataset. The fine-tuning process uses a multi-scale method, gradually incorporating more complex samples. The model is fine-tuned for two tasks in the VEGA dataset: Interleaved Image-Text Comprehension (IITC) and Image-Text Association (ITA). It demonstrates improved performance on both tasks.

**Strengths:**

1. Proposes VEGA, a complex dataset specifically crafted for multi-image understanding, challenging MLLMs with interleaved image and text contexts.

2. Establishes two tasks, Interleaved Image-Text Comprehension (IITC) and Image-Text Association (ITA), setting new benchmarks for interleaved image understanding.

3. Demonstrates that the multi-scale training method effectively improves performance on the ITA task.

**Weaknesses:**

1. Key baseline models such as LLaVA-interleaved and InternVL2 are missing, limiting insight into comparative performance.
2. Similar datasets like SlideVQA, MMLongBench-Doc, DUDE, and DocVQA are not discussed or tested, which could demonstrate the generalization of the fine-tuned model.
3. The ITA task may have limited real-world applicability, as not all text paragraphs have corresponding images. In contexts like research papers, only a subset of text relates to specific images, and filtering irrelevant text often implies inferred relevance between remaining text and images.
4. The font in Figure 1 is too small and should be increased for better readability.

**Questions:**

Other than the weaknesses:

1. Is multi-scale training still effective when not used for multi-task training?
2. In Table 4, what is the model’s performance when trained only on the ITA task with and without multi-scale training?

---

> ### Author Response · Authors · 2024-11-19
>
> # W1: Key baseline models such as LLaVA-interleaved and InternVL2 are missing.
>
> Thank you for your valuable suggestion. It has helped make our experiments more complete and robust.
>
> We have added evaluation results for LLaVA_onevision_7B, Internvl2-8B, and Internvl2-4B on the VEGA dataset. Due to resource and time constraints, we only conducted training on Internvl2-4B. The specific results are as follows:
>
> (Since LLaVA_onevision is the latest SOTA VLM model in the LLaVA series following LLaVA-interleave, we chose to evaluate LLaVA_onevision.)
> | Base model           | Training data | IITC_Rouge | IITC_BLEU | IITC_Acc | IITC_LB. | ITA_3pic | ITA_5pic |
> |-|-|--|-|-|--|--|--|
> | LLaVA_onevision_7B    | -             | 0.367      | 0.115     | 0.241    | 6.34     | 0.172    | 0.028    |
> | Internvl2-8B          | -             | 0.375      | 0.114     | 0.231    | 6.30     | 0.230    | 0.002    |
> | Internvl2-4B          | -             | 0.342      | 0.098     | 0.241    | 5.98     | 0.176    | 0.002    |
> | internvl2-4B          | ITA&IITC      | 0.492      | 0.216     | 0.871    | 6.16     | 0.998    | 0.966    |
>
> As we can see, after training, Internvl2-4B showed a significant improvement in both the IITC and ITA tasks, further validating the effectiveness of the VEGA dataset.
>
> # W2: Similar datasets are not discussed or tested.
>
>
> 1) At the beginning of Section 4.3 in the paper, we validated the impact of the VEGA dataset on the model's general VQA capabilities. We recommend incorporating VEGA into the joint training process alongside other VQA datasets during the SFT phase of MLLMs. This approach can enhance the model's ability to handle interleaved image-text input while maintaining performance on traditional VQA tasks.
>
> 2) Due to time constraints, we only added experimental results for SlideVQA and DocVQA, which are as follows. The results lead to conclusions similar to those in point 1).
> (For MMLongBench-Doc and DUDE, we might need to preprocess the PDFs with OCR first. We're still working on it.)
>
> | Base model           | Training data | IITC_Rouge | IITC_BLEU | IITC_Acc | IITC_LB. | ITA_3pic | ITA_5pic |
> |-|-|-|-|-|-|-|-|
> | LLaVA_onevision_7B    | -             | 0.367      | 0.115     | 0.241    | 6.34     | 0.172    | 0.028    |
> | Internvl2-8B          | -             | 0.375      | 0.114     | 0.231    | 6.30     | 0.230    | 0.002    |
> | Internvl2-4B          | -             | 0.342      | 0.098     | 0.241    | 5.98     | 0.176    | 0.002    |
> | internvl2-4B          | ITA&IITC      | 0.492      | 0.216     | 0.871    | 6.16     | 0.998    | 0.966    |
>
> 3) We believe that when LLMs tackle more difficult tasks, there is often an inherent trade-off. For example, models like LLaMA's math series, code series, and Qwen's math series all sacrifice some general VQA abilities to perform better on specific tasks.
>
> 4) We view SlideVQA, MMLongBench-Doc, DUDE, and DocVQA as fundamentally similar to the traditional VQA tasks validated in Section 4.3, such as MMMU and MME. The input for these datasets generally consists of one or more images with a question, without interleaved image-text input, and the answers are typically simple phrases that do not require the model to indicate a specific image. Therefore, testing on these datasets should yield results consistent with those presented at the beginning of Section 4.3.
>
> # W3: The ITA task may have limited real-world applicability.
> 1) We believe the ITA task has practical applications, such as using RAG-based paragraph segmentation to recommend relevant article sections to users based on images.
>
> 2) its primary purpose is to test and improve the model's ability to associate images when dealing with interleaved image-text inputs. The ITA task serves as a supplementary subtask, trained alongside the IITC task, to help enhance the model's performance in IITC. The focus of this paper is on the IITC task, which has broad real-world use cases.
>
> # W4: About Presentation.
> We have revised the layout of Figure 1 and Figure 2 to improve clarity and presentation, and we apologize for any inconvenience caused to your reading experience.
>
> **For Q1&Q2 please kindly refer to next comment.**

---

> > ### Author Response · Authors · 2024-11-19
> >
> > # Q1&Q2: Is multi-scale training still effective when not used for multi-task training?
> > According to your comment, we add experiments where Qwen-VL-Chat was trained exclusively on the ITA task. The results show that the multi-scale training strategy remains effective even when training is limited to the ITA task.
> >
> > | Base Model    | Multi-scale | ITA 3_pic | ITA 5_pic |
> > |-|-|-|-|
> > | Qwen-VL-Chat  | W           | 0.996     | 0.982     |
> > | Qwen-VL-Chat  | W/O         | 0.166     | 0.006     |
> >
> > The multi-scale training approach aligns with the optimization direction of other contemporary work[1], which focuses on constructing contexts of varying lengths for image inputs. That work found that VLMs exhibit significant fragility as the number of distractor contexts increases. By using training data with different context lengths, we make the training process more robust and stable.
> >
> > Reference:
> > [1]Aditya Sharma , et al. Losing Visual Needles in Image Haystacks: Vision Language Models are Easily Distracted in Short and Long Contexts

---

> > > ### Author Response · Authors · 2024-11-28
> > >
> > > Dear Reviewer,
> > >
> > > We sincerely appreciate your time and effort in reviewing our paper and offering valuable suggestions. As the author-reviewer discussion phase is drawing to a close, we would like to confirm whether our responses have effectively addressed your concerns. We provided responses to your feedback a few days ago, and we hope they have adequately resolved the issues you raised. If you require further clarification or have any additional concerns, please do not hesitate to contact us. We are more than willing to continue our communication with you.
> > >
> > > Best regards, Submission 5626 Authors

---

> > > > ### Comment · Reviewer_qCo5 · 2024-11-28
> > > >
> > > > Thank you for your detailed responses. I have updated my review score to 6.

---

### Official Review · Reviewer_x1Yo · 2024-11-04

**Soundness:** 3
**Presentation:** 3
**Contribution:** 4
**Rating:** 8
**Confidence:** 3

**Summary:**

This paper introduces the Interleaved Image Text Comprehension (IITC) task and establishes the corresponding dataset, VEGA. Compared to existing visual language tasks and datasets, IITC and VEGA have the following unique features: (1) the interleaved images and text input; (2) multiple images input; (3) the task requires the LMM to determine which image is relevant to the question; (4) the input includes image and text information that is irrelevant to the question. Particularly, the latter two features are lacking in previous tasks and datasets.

By benchmarking existing mainstream close-source and open-source MLLMs on VEGA, it was found that the performance of these SOTA  models is not ideal, thus confirming the challenging nature of the IITC task.

Furthermore, this paper also finetunes Qwen-VL-Chat using the train split of VEGA and adopts a multi-scale, multi-task training strategy, laying a solid baseline for subsequent collaborative research.

**Strengths:**

- This paper introduces the Interleaved Image Text Comprehension (IITC) task and creates the corresponding dataset VEGA. IITC and VEGA have the following unique features: (1) the interleaved images and text input; (2) multiple images input; (3) the task requires the LMM to determine which image is relevant to the question; (4) the input includes image and text information that is irrelevant to the question.
These features reflect the normal state of human reading and understanding of image-text materials and are essential for a multimodal AGI system. Existing visual language datasets often lack these features, especially the latter two. Therefore, IITC and VEGA fill this gap to some extent and help to advance the technical progress of the research community in this direction.

- The overall design of the dataset is scientific and reasonable, the construction process is detailed, and the provided statistical information is very convincing. The adopted multi-scale, multi-task training strategy also appears reasonable and credible.

- The main body of the paper is well-structured and written in a clear and concise style, making the article easy to read and understand.

**Weaknesses:**

- All training-related experiments in the paper were conducted solely on the Qwen-VL-Chat model and did not involve other open-source foundation VL models that support interleaved text-image processing. Therefore, the training strategies proposed in this paper and the conclusions from its ablation study have not been sufficiently validated for generalization across other models.
- The article has some issues with formatting and layout, such as the text in some images being too small, which affects normal reading (e.g., Figure 1). Additionally, the layout of the appendix also needs further optimization.

**Questions:**

- During the construction of the VEGA IITC subset, particularly in the answer modification step (see line 305), the authors adjusted the answers from the original dataset (SciGraphQA). Was this modification based on rules (such as keyword matching) or aided by an LLM (e.g., prompt GPT)? Additionally, were the modified answers subjected to quality checks and evaluations, such as through manual sampling inspection and calculating the pass rate?

---

> ### Author Response · Authors · 2024-11-19
>
> # W1: All training-related experiments in the paper were conducted solely on the Qwen-VL-Chat model
> Thank you for your valuable suggestion. It has helped make our experiments more complete and robust.
>
> According to your suggestion, we add experiments using the internvl2 series models. Due to time and resource constraints, we trained a 4B model, and the results are shown in the table below:
> | Base model   | Training data | multi-scale | Rouge | BLEU  | Acc   | L.B. | ITA_3pic | ITA_5pic |
> |--------------|---------------|-------------|-------|-------|-------|------|----------|----------|
> | Internvl2-4B | -             | -           | 0.342 | 0.098 | 0.241 | 5.98 | 0.176    | 0.002    |
> | internvl2-4B | ITA&IITC_4k      | W           | 0.492 | 0.216 | 0.871 | 6.16 | 0.998    | 0.966    |
> | internvl2-4B | ITA&IITC_4k      | W/O         | 0.498 | 0.221 | 0.867 | 6.06 | 0.990    | 0.948    |
> | internvl2-4B | ITA           | W           | -     | -     | -     | -    | 0.998    | 0.984    |
> | internvl2-4B | ITA           | W/O         | -     | -     | -     | -    | 0.996    | 0.972    |
>
> 1) As we can see, after training, Internvl2-4B showed a significant improvement in both the IITC and ITA tasks, further validating the effectiveness of the VEGA dataset.
>
> 2) We can see that the Internvl-4B model already achieved high accuracy on the ITA task without using the multi-scale training strategy, likely because it was trained with interleaved data from OmniCorpus. However, the multi-scale strategy still resulted in slight improvements on both the IITC and ITA tasks, demonstrating the effectiveness of the approach.
>
> 3) While training Qwen-VL-Chat, we encountered poor performance on the ITA task (with only 15.5% accuracy on ITA 3_pic). We believe this is due to the difficulty of the ITA task, leading to training instability. After applying the multi-scale strategy, the performance improved. Training instability may not occur in every model, but when it does, the multi-scale training strategy could be a viable solution.
>
> # W2: About Presentation
> We have revised the layout of Figure 1 and Figure 2 to improve clarity and presentation, and we apologize for any inconvenience caused to your reading experience.
>
> # Q1: About the answer modification step
> > Was this modification based on rules (such as keyword matching) or aided by an LLM (e.g., prompt GPT)?
>
> In the answer modification step, we used a keyword matching approach. We first reviewed over 700 sample answers from the test set and developed corresponding keyword matching rules.
>
> > Additionally, were the modified answers subjected to quality checks and evaluations, such as through manual sampling inspection and calculating the pass rate?
>
> After performing the answer modification, we manually reviewed all of the answers in the test set. In the VEGA4k test set, only two answers were incorrectly modified (an error rate of about 0.3%), and these errors have been removed. As for the training set, due to its larger size, we did not conduct additional manual filtering. Given that the proportion of erroneous data is very small, we consider it as noise during training.

---

### Official Review · Reviewer_QKxm · 2024-11-05

**Soundness:** 3
**Presentation:** 2
**Contribution:** 4
**Rating:** 6
**Confidence:** 4

**Summary:**

This paper introduces a novel task, Interleaved Image-Text Comprehension (IITC), which evaluates models' capabilities to handle complex interactions in lengthy multimodal contexts. The authors present the VEGA dataset specifically designed for the IITC task, featuring up to 8,000 tokens and 8 images. The paper also benchmarks several models on this task, including proprietary and open-source options, and demonstrates the benefits of fine-tuning Qwen-VL-Chat for IITC. The results show that even state-of-the-art models face challenges in this setting, highlighting the dataset's rigor and the task's demands.

**Strengths:**

The introduction of a challenging new task (IITC) with extensive multimodal input length sets a unique standard in evaluating models' interleaved image-text comprehension. The proposed VEGA dataset is extensive, comprising 295k high-quality, multi-turn entries, with a curated test set of 700 questions to assess models' nuanced comprehension abilities.

The benchmark evaluates both proprietary and open-source models, showing that current models, even at their best, are still only moderately successful in IITC tasks. Fine-tuning Qwen-VL-Chat leads to meaningful improvements, establishing a strong baseline for future models.

The dataset structure, including partitions for 4k and 8k token contexts, is well-conceived and allows for detailed performance analysis. Including specific references to image indices in responses (e.g., '[Picture 2](Figure. 3)') is a thoughtful design choice, aiding interpretability in output.

**Weaknesses:**

Presentation: Figures, particularly Figure 1 and Figure 2, are too small, making the text nearly unreadable without significant zoom. This compromises the clarity and accessibility of the paper's visual aids.

Test Set Generalizability: The test set, created through a process similar to the training data generation, may inadvertently advantage trained models by being in-distribution. Introducing an additional test set, created independently or using different datasets, could strengthen the evaluation by ensuring generalization beyond the training data distribution.

**Questions:**

Would you consider adding an out-of-distribution test set to further evaluate generalization capabilities?

---

> ### Author Response · Authors · 2024-11-19
>
> # W1: About Presentation
> We have revised the layout of Figure 1 and Figure 2 to improve clarity and presentation, and we apologize for any inconvenience caused to your reading experience.
>
> # W2&Q1: About Test Set Generalizability
> Thank you very much for your suggestion. We have also considered other application scenarios for interleaved image-text input, such as instruction manual comprehension, Q&A for guides and tutorials, and textbook-based question answering. We plan to construct test sets based on these real-world applications to better evaluate the model's ability to handle interleaved image-text input.

---

> > ### Comment · Reviewer_QKxm · 2024-11-21
> >
> > Thank you. Do you plan to have the new tests ready for the camera ready version ?

---

> > > ### Author Response · Authors · 2024-11-22
> > >
> > > With nearly three months remaining until the camera-ready deadline, we believe we can gather a sufficient dataset for evaluation. We have collected a small sample of mobile phone manuals, which have proven effective in evaluating existing models' performance on IITC tasks and their multi-image processing capabilities. Before the camera-ready submission, we will assess data licensing requirements and dataset size considerations to prepare for the dataset's public release.

---

> > > > ### Comment · Reviewer_QKxm · 2024-11-26
> > > >
> > > > Thanks for the response. The point of missing out-of-distribution test set is a large weakness in my opinion. I will retain my original score. If all raters lean to accept, it's fine with me. But I think an additional test set will really improve the paper.

---

> > > > > ### Author Response · Authors · 2024-11-27
> > > > >
> > > > > Thank you for your valuable feedback. In fact, most existing datasets and benchmarks (when a training set is provided) are typically split into train, dev, and test sets from in-distribution data. For example, benchmarks like OCRVQA provide both test data and a large amount of in-distribution training data.
> > > > >
> > > > > We understand your concerns and fully agree that adding test data from other interleaved image-text scenarios could enhance the reliability of the IITC task evaluation. We are currently working on collecting new test data and will conduct experiments by track (such as reading comprehension in papers, textbooks, user manuals, and online posts) to ensure fairness in evaluation.
> > > > >
> > > > > We will complete this work before the camera-ready submission.

---

### Official Review · Reviewer_Mss2 · 2024-11-08

**Soundness:** 3
**Presentation:** 2
**Contribution:** 3
**Rating:** 6
**Confidence:** 4

**Summary:**

Long-context interleaved image-text comprehension is an important skill required to read scientific papers, for example. Humans can do this well, but characterization of this ability is understudied. This paper introduces a multimodal task called Interleaved-Image-Text Comprehension which requires answering a question using interleaved image and text and grounding the answer in one of the in-context images. The authors develop a dataset called VEGA to evaluate models on this task and auxilliary simpler task called image-text association. They evaluate multiple frontier LLMs and some open LLMs on the task, and develop a training strategy which they show improves performance on this task.

**Strengths:**

- There are no resources (as far as I know of) to study and characterize _long-context_ interleaved image-text comprehension or empirical studies of this comprehension ability. This is one of the first.
- The experimental evaluation is broad, covering multiple vision language models including a mix of closed source LLMs and open source LLMs.
- I especially appreciate the honest of the authors in showing that training on Vega _does not_ in general improve performance on other VQA tasks.

**Weaknesses:**

- The construction method of the dataset seems a bit artificial.
- I don't see analyses of how the number of images / context length affects accuracy. In general, I feel like important analyses are not present. There are no "slices" in the dataset to sanity check that expected properties hold. For example, I would expect that as context increases, accuracy falls. If I did not see this, I would be suspicious of the construction method. This is just an example. My main point is that the dataset and how the properties of the dataset affect evaluation are not analyzed enough or perhaps not explained in the text.
- I am not convinced by the proposed training method. It is not really motivated or explained well in the text, and there is no discussion (as far as I can tell, but it is very possible I have missed something) on how it relates to other similarly named methods in the literature for "multi-scale training".

**Questions:**

Please see the weaknesses section. Most of the weaknesses are minor. The weakness I care about and the one where an author response would cause me to increase my rating is "analyses". What analyses have the authors done on how properties of the dataset (e.g. number of images, context length, etc) affect the performance of MLLMs? Also, can the authors discuss why there are multiple evaluation metrics and what each of them signify?

---

> ### Author Response · Authors · 2024-11-19
>
> # W1: The construction method of the dataset seems a bit artificial.
> During the dataset construction process, one somewhat artificial step is the modification of the answers, where we add [Picture i](Figure j). We hope that the model can learn to use such standardized special tokens to refer to the images mentioned in its responses. This ability is particularly meaningful when the model handles interleaved image-text input.
>
> Following your suggestions, we revised the VEGA dataset by replacing [Picture i] (Figure j) with more natural language, such as “the first image” or “the second image.” We tested these changes on both Qwen-VL-Chat and Qwen-MAX, and observed similar results:
> | Model        | IITC 4k_nature_language |            |            | IITC 4k      |            |            |
> |-|-|-|-|-|-|-|
> |              | Rouge-L                 | BLEU       | Pic_Acc    | Rouge-L      | BLEU       | Pic_Acc    |
> | Qwen-max     | 0.359                   | 0.116      | 0.72       | 0.356        | 0.107      | 0.684      |
> | Qwen-VL-Chat | 0.331                   | 0.083      | 0.002      | 0.323        | 0.093      | 0.002      |
>
> This shows that using natural language produces similar results to [Picture i] (Figure j). However, using special tokens like [Picture i] (Figure j) is more consistent and standardized.
>
> # W2: About the analysis of dataset.
> **Number of images:**
>
> In Figure 5 of the paper, we present the relationship between the number of images and the accuracy of image association in the IITC task. From the figure, we can see the following:
> 1. The image association accuracy of the VEGA-base-4k model decreases as the number of images increases.
> 2. For the other closed-source models, there is also a general negative correlation between the number of images and image association accuracy.
> The increase in the number of images makes image selection in the IITC task more challenging.
>
> **Context length:**
>
> | model       | 0-1k | 1-2k | 2-3k | 3-4k | 4-5k | 5-6k | 6-7k | 7-8k |
> |-------------|------|------|------|------|------|------|------|------|
> | VEGA-4k     | 1.00 | 0.886| 0.850| 0.859| 0.779| 0.799| 0.771| 0.857|
> | internvl-1.5| 0.22 | 0.409| 0.344| 0.352| 0.256| 0.313| 0.252| 0.357|
> | Gemini-1.0  | 1.00 | 0.726| 0.617| 0.667| 0.530| 0.448| 0.471| 0.607|
> | Gemini-1.5  | 1.00 | 0.770| 0.747| 0.855| 0.738| 0.726| 0.656| 0.929|
>
> We have supplemented the analysis with the relationship between token length and image accuracy. Details can be found in the table above:
>
> 1) Statistically, there is a general negative correlation between image accuracy and token length.
>
> 2)  Due to the uneven distribution of token lengths in the test set (see Figure 4 of the paper), there is a limited amount of test data in the 0-1k and 7-8k ranges (with only 9 and 28 samples, respectively), which may lead to some margin of error in these intervals.
>
> 3) As shown in Table 2 of the paper, for all models, the image accuracy in IITC 4k is higher than in IITC 8k, further supporting the negative correlation between accuracy and token length.
>
> The increase in context length introduces more redundant information, making image selection more challenging.
>
> **Image Similarity and Text Complexity:**
>
> Many of the input images in the VEGA dataset are highly similar, and the accompanying text is complex and often misleading, making the data more challenging. For example, in many physics papers, the charts have very similar structures, sometimes differing only in the units on the axes. This requires the model to have strong OCR capabilities to differentiate between them. In some cases, the text that is highly relevant to the question refers to two or more images simultaneously, which can easily lead to incorrect image selection by the model.
>
> **Question Complexity:**
>
> Many of the questions in VEGA are quite complex. For instance, a question like "How does the graph compare the performance of the Q-SPRT and RLT-SPRT algorithms?" requires not only describing the content of the image but also analyzing it in combination with the text. Such questions are more challenging compared to traditional VQA tasks, which may only involve simple OCR information extraction or answering a short phrase.
>
> For more cases in VEGA， please refer to Appendix. C.
>
>
> For W3, Q1&Q2, please kindly refer to next comment.

---

> > ### Author Response · Authors · 2024-11-19
> >
> > # W3:  About training method.
> > > I am not convinced by the proposed training method. It is not really motivated or explained well in the text.
> >
> > **Motivation:** The motivation behind our proposed multi-scale training strategy comes from the observation that Qwen-VL-Chat performed poorly on the ITA task (with 15.5% accuracy on ITA_3Pic), which was essentially equivalent to random guessing. We believe this is due to the inherent difficulty of the ITA task, where the input segments often include a large amount of text unrelated to the images. As a result, the model learned the output format but failed to grasp the correspondence between the images and the segments.
> >
> > **Explain:** To address this issue, we proposed a multi-scale training strategy. We introduced two additional scales, replacing the segments in the ITA task with the first_mention (the first sentence in the text that references the image) and the image caption, both of which have a stronger correlation with the image. This lowers the difficulty of the task and allows for joint training, making it easier for the model to understand that the goal is to find the correspondence between the image and the text. For related ablation studies, please refer to Table 4 in the paper. After incorporating multi-scale training, the model’s performance on the ITA task improved significantly, with improvements also seen in the IITC task.
> >
> > >There is no discussion on how it relates to other similarly named methods in the literature for "multi-scale training".
> >
> > **Discussion on similar method:** The multi-scale training approach aligns with the optimization direction of other contemporary work[1], which focuses on constructing contexts of varying lengths for image inputs. That work found that VLMs exhibit significant fragility as the number of distractor contexts increases. By using training data with different context lengths, we make the training process more robust and stable.
> >
> > # Q1:About the analyses of the dataset.
> > Please kindly refer to W2.
> > # Q2:  Can the authors discuss why there are multiple evaluation metrics and what each of them signify?
> > In the IITC task, we use four metrics to evaluate the task from multiple dimensions: Rouge-L, BLEU, LLaVA-Bench scores, and accuracy (Acc).
> > 1. Rouge-L and BLEU are traditional metrics used to assess the quality of generated text in NLP tasks. Both work by comparing the similarity between the generated text and a reference text to measure the quality of the output. However, their limitation is that these phrase-based matching metrics often fail to capture semantic similarity.
> > 2. The LLaVA-Bench scores are an evaluation metric based on LLM. We used Llama 3.1-70B to score the model's responses. LLM-based evaluation methods help compensate for the semantic shortcomings of Rouge-L and BLEU. By combining these three metrics, we can better assess the quality of the model's responses.
> > 3. Accuracy (Acc) is used to measure the model's ability to associate images correctly when dealing with interleaved image-text input.
> >
> > Reference:
> > [1]Aditya Sharma, , et al. Losing Visual Needles in Image Haystacks: Vision Language Models are Easily Distracted in Short and Long Contexts

---

> > > ### Comment · Reviewer_Mss2 · 2024-11-21
> > >
> > > Thank you for the reply, authors.
> > >
> > > > During the dataset construction process, one somewhat artificial step is the modification of the answers, where we add [Picture i](Figure j).
> > >
> > > I don't think the modification of the answers is artificial, I think the "Context Construction" (L260-265) is artificial.
> > >
> > > > "The second expands the text-image sequence from the original paper..."
> > >
> > > I don't see details of how this expansion is done. But this seems critical to the validity of the dataset. In the same paragraph you also mention that there are two approaches to crafting the long texts used in the dataset. How was the approach you eventually settled on chosen? Why was it chosen?
> > >
> > > I'm not looking for more experiments to answer these questions. I am looking for some explanations, maybe some intuition for why an approach or the other was chosen and why. This is primarily a dataset paper and these decisions should be explained. I don't mean explain them to me in the rebuttal either, I mean you should add information to the Appendix explaining these things.
> > >
> > > **Can you fix Figure 4 and Figure 5?** They don't look like ICLR quality figures. The fonts are mismatched from the rest of the paper and Figure 5 looks _very_ low quality.
> > >
> > > **Can you add your explanation of the motivation and intuition behind the multi-scale training strategy somewhere in the paper, maybe the Appendix?** If you have already added it, can you point me to it. Otherwise, it feels like this strategy came out of nowhere.
> > >
> > > Overall, I like the paper and _want_ to raise my score. I am happy with the experiments as well but as someone who would like to use your dataset, I want to have some confidence _from reading the paper alone, without having to look at the data_ that the benchmark was methodically constructed and each decision was taken carefully and not haphazardly. You should document these decisions somewhere (even briefly, in the Appendix is fine). Just describe your reasons behind why you did what you did so that others know.

---

> > > > ### Author Response · Authors · 2024-11-21
> > > >
> > > > We greatly appreciate your positive feedback and valuable suggestions, which have helped us improve the clarity of our paper.
> > > >
> > > > # Q1: About Context Construction
> > > > > I don't see details of how this expansion is done. But this seems critical to the validity of the dataset. In the same paragraph you also mention that there are two approaches to crafting the long texts used in the dataset. How was the approach you eventually settled on chosen?
> > > >
> > > > We experimented with two approaches to construct the context for IITC tasks.
> > > >
> > > > **Approach 1**: We defined an image-text pair as an image and its first mention (the first few sentences that reference the image) in the article. Multiple image-text pairs from different articles were selected to form the context for IITC tasks. We then formulated questions about one of these image-text pairs, requiring the model to specify which image it referenced in its response.
> > > >
> > > > **Approach 2**: We constructed the context from a single article. Starting with the first mention of the target image, we incrementally added sentences from the TEX file either upward or downward until reaching a predetermined context length threshold. The direction (up or down) was randomly determined to avoid having the first mention consistently appear in the middle of the context. Throughout this process, we preserved the spatial relationships between images and text as specified in the TEX file. We then posed questions about the target image and its associated text, requiring the model to specify which image it was referencing when providing answers.
> > > >
> > > > > Why was it chosen?
> > > >
> > > > We chose Approach 2 for context construction based on the following considerations:
> > > >
> > > > 1)  Approach 2 better simulates real-world applications. When MLLMs perform long document understanding with RAG, the model input typically consists of relevant text segments, where retrieved textual and visual contexts are both coherent and potentially ambiguous. Approach 2 mimics this retrieval-based scenario, training the model to be robust against confounding textual and visual contexts. In contrast, Approach 1 produces highly distinct segments with lower contextual complexity, which diminishes the IITC task difficulty. This is validated by our ablation experiments in Section 4.3 (line 516), which demonstrate superior training outcomes using Approach 2.
> > > >
> > > > 2) Existing pre-trained interleaved image-text datasets, such as MINT-1T[1], also source images and text from the same article, establishing coarse-grained image-text alignment. Approach 2's IITC task construction pushes the model further by requiring it to locate relevant segments and perform more fine-grained comprehension. Our dataset, constructed through Approach 2, enables training with coherent and progressive image-text interleaving.
> > > >
> > > > 3) While Approach 1 is less resource-intensive and similar results could be achieved by concatenating normal QA datasets, we opted for Approach 2 to enhance the diversity of MLLM comprehension capabilities.
> > > >
> > > > >  I mean you should add information to the Appendix explaining these things.
> > > >
> > > > Thank you for your valuable suggestions, which have helped improve the presentation of our paper. We will add more detailed information about dataset design and model training in the Appendix, including specifics about Context Construction and training strategies.
> > > >
> > > > # Q2: Can you fix Figure 4 and Figure 5?
> > > > We will improve the presentation of Figure 4 and Figure 5.
> > > >
> > > > # Q3: Can you add your explanation of the motivation and intuition behind the multi-scale training strategy somewhere in the paper, maybe the Appendix?
> > > >
> > > > > If you have already added it, can you point me to it. Otherwise, it feels like this strategy came out of nowhere.
> > > >
> > > > We provided an explanation of the multi-scale training strategy in lines 319-325 of the paper. We apologize if formatting or presentation issues made this section difficult to follow.
> > > >
> > > > We will include additional details about dataset design and model training in the Appendix. As you mentioned, both the context construction and training strategy will be elaborated in greater detail there. Thank you for your valuable suggestions.

---

> ### Comment · Reviewer_Mss2 · 2024-11-22
>
> > Despite the task’s complexity, leading-edge methods like Qwen-MAX Qwen-VL Team (2024) and Gemini-1.0-pro Team et al. (2024) exhibit only moderate success, particularly as the volume of pairs escalates, as evidenced in Table 2. At the same time, we find that direct training with expanded image-text pairs is insufficient for the model to master text-image associations. To mitigate the learning challenge, we implement a multi-scale training strategy in the ITA task. We design three textual scales for the training set, corresponding to the image caption Ci, the first mention paragraph Mi, and the expanded context paragraphs Ei. We also design two image quantity scales, with sets of three and five images.
>
> These lines from the paper are not a good explanation. I don't see any intuition for why the multi-scale strategy works or is a good baseline.
>
> > We will improve the presentation of Figure 4 and Figure 5.
>
> There are still several days remaining in the discussion period and you can upload a revised PDF. Please update these figures. If you update these figures, I will feel comfortable raising my score and trust that you will make the other changes as well.

---

> ### Comment · Reviewer_Mss2 · 2024-11-22
>
> Regarding the multi-scale training parts, I don't feel it is explained well, but it is not the main contribution of the paper. Don't worry about it.

---

> > ### Author Response · Authors · 2024-11-22
> >
> > Thank you for your reply. We have revised Figures 4 and 5 in the paper according to your suggestions, unifying the style, font, and color of the figures. Additionally, based on feedback from other reviewers, we have also updated Figure 1 to make sure the figure is clear without needing to zoom in. All these changes can be found in the revised PDF.
> >
> > We are ready to incorporate any further suggestions you may have to improve our work.

---

> > > ### Comment · Reviewer_Mss2 · 2024-11-23
> > >
> > > Thank you, I have raised my score. Good luck!

---

> > > > ### Author Response · Authors · 2024-11-24
> > > >
> > > > Dear Reviewer Mss2,
> > > >
> > > > We sincerely appreciate your recognition of our efforts and the time you dedicated to reviewing our paper.
> > > >
> > > > Respectfully,
> > > >
> > > > Authors of Paper 5626

---

### Author Response · Authors · 2024-11-19

# Response to All Reviewers
We would like to thank all the reviewers for their thorough reviews and highly valuable feedback and suggestions. We are excited to hear that the reviewers found our work meaningful [R1, R2, R3, R4], well-designed [R2 R3], and experimentally rigorous [R1, R2].

We also appreciate the suggestions provided, which have improved our work. We noticed that many reviewers pointed out the issue with the font size in the images, and we have revised the layout of Figure 1 and Figure 2 to improve clarity and presentation, and we apologize for any inconvenience caused to your reading experience.

# Supplementary experiments
1. Added evaluation results for Internvl2-4B, Internvl2-8B, and LLaVA-OneVision-7B on the VEGA dataset.
2. Provided analysis on the relationship between token length and image accuracy.
3. Trained Internvl2-4B on the VEGA dataset.
4. Included the test results of Qwen-VL-Chat on SlideVQA and DocVQA, both before and after training.
5. Added test results for Qwen-VL-Chat trained exclusively on the ITA task.
6. Tested the use of natural language to replace [Picture n], such as "first image" instead of [Picture 1], to improve the generalizability of evaluations on the VEGA dataset.

---

### Meta-Review · Area_Chair_Cvhm · 2024-12-20

**Metareview:**

All reviewers gave positive evaluations. The authors introduce a new task called Interleaved Image-Text Comprehension and provide the corresponding VEGA dataset to support this task. This task challenges models to understand complex interleaved image-text content, a valuable contribution to the vision-language research community. The proposed multi-scale training strategy effectively improves model performance. The authors should clarify certain aspects of the dataset creation and training methods to enhance the paper's clarity and comprehensibility. Overall, this paper is expected to inspire further research in this field.

**Additional Comments On Reviewer Discussion:**

During the discussion, several concerns were raised. Reviewer Mss2 questioned the dataset's construction, suggesting it appeared artificial, and sought clearer explanations of the multi-scale training strategy. The authors responded by justifying their dataset design as representative of real-world scenarios and providing detailed clarifications of the training strategy. Reviewer x1Yo recommended testing the training strategies on additional models to demonstrate generalizability, which the authors addressed by conducting experiments across various models. Reviewer qCo5 raised concerns about the real-world applicability of the ITA task and the effectiveness of the multi-scale training strategy without multi-task training. In response, the authors demonstrated the practical relevance of the ITA task and showed that their training strategy remains effective even when applied solely to this task. Having thoroughly addressed these concerns and considering the reviewers' positive evaluations, this paper makes valuable contributions to the field of vision-language research.

---

### Decision · Program_Chairs · 2025-01-22

Accept (Poster)